

# Synoptic processes of winter precipitation in the Upper Indus Basin

Jean-Philippe Baudouin[1,2], Michael Herzog[2], and Cameron A. Petrie[3]

[1]Department of Environmental Physics, Heidelberg University, Germany
[2]Department of Geography, University of Cambridge, United Kingdom
[3]Department of Archaeology, University of Cambridge, United Kingdom

**Correspondence:** Jean-Philippe Baudouin (baudouin.jeanphilippe@gmail.com)

**Abstract.**

Precipitation in the Upper Indus Basin is triggered by cross-barrier moisture transport. Winter precipitation events are particularly active in this region and are driven by an approaching upper troposphere Western Disturbance. Here statistical tools are used to decompose the winter precipitation timeseries into a wind and a moisture contribution. The relationship between each contribution and the Western Disturbances are investigated. We find that the wind contribution is not only related to the intensity of the upper troposphere disturbances but also to their thermal structure through baroclinic processes. Particularly, a short-lived baroclinic interaction between the Western Disturbance and the lower altitude cross-barrier flow occurs due to the shape of the relief. This interaction explains both the high activity of Western Disturbances in the area, as well as their quick decay as they move further east. We also revealed the existence of a moisture pathway from the Red Sea, to the Persian Gulf and the north of the Arabian Sea. A Western Disturbance strengthens this flow and steers it towards the Upper Indus Plain, particularly if it originates from a more southern latitude. In cases where the disturbance originates from the north-west, its impact on the moisture flow is limited, since the advected continental dry air drastically limits the precipitation output. The study offers a conceptual framework to study the synoptic activity of Western Disturbances as well as key parameters that explain their precipitation output. This can be used to investigate meso-scale processes or intra-seasonal to inter-annual synoptic activity.

## 1 Introduction

The Upper Indus River Basin (UIB) is a mountainous region that differs from the rest of the Indian subcontinent for the large amount of precipitation it receives outside of the summer monsoon season (56% or 505mm between October and May for ERA5 on the period 1979 and 2018, Baudouin et al., 2020b). Much of this precipitation falls as snow at altitude during the coldest part of the season (Hewitt, 2011; Dahri et al., 2018; Baudouin et al., 2020b). The precipitation and the snowmelt later in the season are key for mitigating the seasonal drought that occurs for most of South Asia before the arrival of the summer monsoon (Singh et al., 2011; Dimri et al., 2015; Rana et al., 2015).

In either winter or summer, precipitation in the UIB is triggered by the forced up-lift of a moisture transport perpendicular to the mountain ranges (Baudouin et al., 2020a). However, the synoptic drivers of cross-barrier transport differ between summer and winter. The winter drivers have been discussed since the mid-twentieth century (Malurkar, 1947; Mull and Desai, 1947,





cf. in Dimri et al. 2015). Winter precipitation events in the UIB are related to the passing of extra-tropical, synoptic-scale, disturbances often originating from the Mediterranean Sea, the Black Sea, or the Caspian Sea, and referred to as Western Disturbances (WDs, Dimri et al., 2015; Hunt et al., 2018a). Many numerical case studies have investigated the characteristics of WD's and their interaction with the relief (e.g. Dimri, 2004; Dimri and Niyogi, 2013; Thomas et al., 2018; Krishnan et al.,
2018). More recently, exhaustive tracking analysis have been carried out on generalised previous results (Syed et al., 2010; Cannon et al., 2016; Hunt et al., 2018a). Despite the abundant interest, a precise and non case-specific understanding of the relationship between WD characteristics and precipitation variability is still lacking.

Typically up to six or seven WDs occur per month (Hunt et al., 2018a), although lower frequencies have been observed (Cannon et al., 2015; Dimri, 2013), probably depending on threshold intensity. A WD is characterised by a maxima of vorticity
or a minimum in geopotential height near the tropopause between $325\,\mathrm{hPa}$ (Hunt et al., 2018a) and $200\,\mathrm{hPa}$ (Midhuna et al., 2020). The cyclonic circulation around the WDs interacts with the relief to trigger precipitation (Baudouin et al., 2020a). Generally, a stronger cyclonic circulation induces more precipitation (Hunt et al., 2018a). Yet, the convergence triggering precipitation occurs at a much lower altitude, around $700\,\mathrm{hPa}$ and below (Hunt et al., 2018a; Baudouin et al., 2020a), which suggests that the downward propagation of the cyclonic circulation is key to producing precipitation. Alternatively, Dimri and
Chevuturi (2014) have proposed that the WDs interact with a pre-existing low-level cyclonic circulation located over the Thar desert, reminiscent of the heat low present in the area during summer (Bollasina and Nigam, 2011). The baroclinic interaction between an upper- and a lower-level trough is known to be a key process in the growth of extra-tropical disturbances (Malardel, 2005). Furthermore, Hunt et al. (2021) show that interactions between WDs and tropical depression exist, although these are not common in winter.

The upper troposphere disturbance characterising a WD is embedded in the Subtropical Westerly Jet (SWJ, Dimri et al., 2015; Hunt et al., 2018a). The SWJ characterises the Northern edge of the Hadley circulation (Krishnamurti, 1961). In Asia, the Tibetan Plateau disrupts the SWJ. The jet oscillates between two stable states: one north of the Tibetan Plateau, always present in summer, and one south of it, reached only in winter (Schiemann et al., 2009). Furthermore, in winter, the SWJ is also split in two climatological jet streaks of higher intensity (Krishnamurti, 1961; Schiemann et al., 2009): one over the Arabian
Peninsula (Arabian Jet, Yang et al., 2004; de Vries et al., 2016), and the other over East Asia (East Asian Jet, Xueyuan and Yaocun, 2005). It has been argued that the position and strength of the SWJ influences WD intensity at the intra-seasonal and inter-annual scale (Filippi et al., 2014; Dimri et al., 2015; Hunt et al., 2018a; Ahmed et al., 2019). These studies suggest that the higher kinetic energy in the SWJ is able to fuel the development of WDs as for other baroclinic waves. Yet, Hunt et al. (2018a) have mentioned that WDs are immature baroclinic waves that differ from their mature counter-part in the Atlantic or
Pacific Ocean: the WDs remain in a nascent state without low-level warm core or frontal activities. The coupling between the SWJ, WDs and the relief needs to be better characterised as they greatly influence vertical velocities, in part through baroclinic processes, and thus precipitation. In particular, this coupling could explain why WDs are particularly active in the UIB.

While wind is the most important parameter to explain precipitation synoptic variability, moisture content modulates the strength of the relationship (Baudouin et al., 2020a). Previous studies have investigated moisture transport in the context of
winter precipitation in the UIB using moisture flux (Dimri, 2007; Syed et al., 2010; Filippi et al., 2014; Hunt et al., 2018a)





or back trajectories (Jeelani et al., 2018; Hunt et al., 2018b; Boschi and Lucarini, 2019). Yet, some uncertainty remains about the moisture sources for precipitation and its pathways. The Arabian Sea is often suggested as the primary source of moisture (Dimri, 2007; Filippi et al., 2014; Hunt et al., 2018b). Less certain is the input of moisture from the Mediterranean Sea, as it is sometimes suggested that it is the origin of the most intense WDs that reach the UIB (e.g. Filippi et al., 2014; Dimri et al., 2015). Occasionally, the Red Sea (Dimri, 2007; Filippi et al., 2014), the Caspian Sea (Syed et al., 2010; Dimri and Niyogi, 2013) or even the Atlantic Ocean (Dimri et al., 2015) are also mentioned. The moisture pathway is certainly affected by the passing of a WD, through the deformation of the low-level wind field the WD imposes (Baudouin et al., 2020a). Yet, the reason for moisture variability at the synoptic scale has not been investigated extensively.

The objective of this study is to understand the synoptic variability of winter precipitation in the UIB and how this relates to various characteristics of WDs. The analysis makes use of reanalysis data and extends statistical tools developed in Baudouin et al. (2020a, section 3). The analysis particularly focuses on understanding the origin of cross-barrier wind variability and moisture sources.

## 2  Season, data and study area

The timeseries of precipitation considered here is defined by a 3-hourly average over the UIB (cf. black contour in Figure 1-A). This study area is the same as the one used in Baudouin et al. (2020a) and Baudouin et al. (2020b). Both precipitation and atmospheric variables are derived from ERA5 reanalysis, at a 3-hourly intervals and 0.5° resolution, over the 40-year period 1979-2018. ERA5 proved to provide a good representation of precipitation variability (Baudouin et al., 2020b) and synoptic processes (Baudouin et al., 2020a). ERA5 data provide extrapolated values on pressure levels below the model surface which will be excluded from the analysis. Consequently, grid points where the minimum geopotential is above the model surface were deselected[1].

All analyses are performed over an extended winter season that spans from October to May. Similar studies on WDs generally considered a shorter winter period to avoid the dry intermediate seasons (e.g. December-February in Midhuna et al. 2020 or December-April in Hunt et al. 2018b). However, it is possible to demonstrate that large precipitation events during these intermediate seasons are driven by the same synoptic disturbances as in winter (cf. Baudouin, 2020, Figure 4.18). This selection also increases the diversity of background conditions that are considered.

---

[1]This is in contrast to Baudouin et al. (2020a), which used an arbitrary pressure threshold



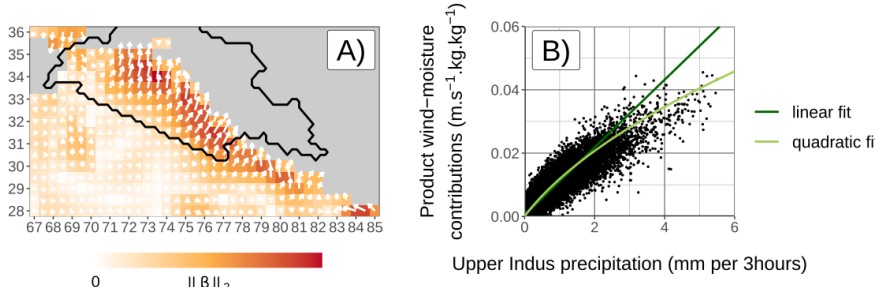

**Figure 1.** Results of the wintertime regression of precipitation with meridional and zonal moisture transport at $700\,\mathrm{hPa}$. Precipitation is defined as the 3-hourly averaged over the UIB (black contour Panel A). Panel A shows the coefficients associated with each predictor. The arrows indicate the most efficient direction of moisture transport, while the length and the colour of the arrows indicate the total weight of moisture transport (i.e. the Euclidean norm of the coefficients from both zonal and meridional components). The exact values of the coefficients are not interpretable and are therefore not indicated. Panel B is the scatter plot of the product of wind and moisture contributions against precipitation, with the green lines representing two types of fit.

## 3 Methods

### 3.1 Computing wind and moisture contributions to precipitation

Baudouin et al. (2020a) demonstrated the ability of principal component (PC) regression to analyse the precipitation variability in terms of moisture transport, defined as the product of wind and specific humidity. A similar method is used here, with small differences, to both simplify and test the robustness of the method. The reasons for each of the changes in the approach and their impact on the coefficient of determination (or explained variance) of the regression is further detailed in Baudouin (2020).

The method uses moisture transport at $700\,\mathrm{hPa}$ to predict 3-hourly precipitation between October and May. Moisture transport is considered over an area between 18 and 36°N and between 67 and 85°E, for a resolution of 0.5° (cf. the extent and resolution in Figure 1-A). The PC analysis is applied on centred timeseries of moisture transport and considers both meridional and zonal components simultaneously. The first 46 PCs are selected for the regression.

The prediction of precipitation is a linear combination of the PCs of moisture transport. However, since precipitation can only be positive, negative predictions are fixed to 0. This condition is directly applied to the statistical model of the regression, which is solved using an iterative optimisation method (Brent, 1973). Finally, a condition to correct any bias is also added to the statistical model. The coefficient of regression of this regression is $R^2 = 0.878$ which is very close to the one found in Baudouin et al. (2020a, $R^2 = 0.832$).

Figure 1-A shows the coefficients of the regression: the arrows are constructed from the coefficients of each component, while the colour is proportional to the length of the arrow. The arrows highlight the moisture convergence zone along the Himalayan foothills that triggers precipitation (cf. Baudouin et al., 2020a).





As in Baudouin et al. (2020a), the prediction is split between the contribution of wind and moisture, using a weighted spatial averaging. The wind contribution (labelled hereafter $W700$) is computed by multiplying the time series of meridional and zonal wind with the respective meridional and zonal regression coefficients and summing the result. $W700$ is interpreted as the cross-barrier wind effect on the precipitation. For the moisture contribution (labelled hereafter $Q700$), the time series at each location are weighted with the euclidean norm of the coefficients of both meridional and zonal moisture transport (i.e. the colour scheme used in Figure 1-A). After fixing negative values of $W700$ to 0, the product of $Q700$ and $W700$ has an $R^2$ of 0.844 with precipitation, close to the $R^2$ of the regression with moisture transport (0.878), despite the spatial averaging. Note that the intercept of the regression with moisture transport equals $0.14\,\mathrm{mm}\cdot 3\mathrm{hours}^{-1}$, but is not taken into account when considering the product of $Q700$ and $W700$.

Figure 1-B shows the product of $Q700$ and $W700$, against precipitation. A non-linearity can be seen as precipitation increases quicker than the product. This non-linearity is not present when including moisture transport at $850\,\mathrm{hPa}$ in the PC regression (cf. Baudouin et al., 2020a). This behaviour can be explained by the fact that moisture transport at lower altitude is not proportional to that of higher altitude and instead quickly increases once strong moisture transport is already present at $700\,\mathrm{hPa}$ (Baudouin et al., 2020a). An ad hoc quadratic fit of the precipitation with the product of $Q700$ and $W700$ captures the non-linearity and slightly increases the $R^2$ from 0.844 to 0.853. Hence, the product of $Q700$ and $W700$ is a good predictor of precipitation and each contribution will be investigated separately in the following sections.

## 3.2 Relating cross-barrier wind to Western Disturbances

The second methodological step consists in using $W700$ to investigate the link between the cross-barrier wind intensity and various WDs characteristics.

First, a qualitative analysis is proposed using a composite. The composite is defined as the average over atmospheric fields of the time steps with the 10% highest value of $W700$ (i.e. 3-hourly $W700$ above $2.68\,\mathrm{m\,s}^{-1}$). This selection of time steps accounts for about 50% of the precipitation between October and May. To remove the influence of the seasonality, the anomalies of the atmospheric fields are computed by removing the four first harmonics of the seasonal cycle. This approach ensures, for example, that the monthly mean anomalies of geopotential height at $300\,\mathrm{hPa}$ are below $100\,\mathrm{m}^2\,\mathrm{s}^{-2}$. Various pressure-level fields are investigated: the geopotential height anomaly at $300\,\mathrm{hPa}$ and the anomaly of geopotential thickness between $500\,\mathrm{hPa}$ and $300\,\mathrm{hPa}$ (Figure 2); the wind speed and wind speed anomaly at $250\,\mathrm{hPa}$ (Figure 4); the precipitable water anomaly and the absolute water vapour flux (Figure 9-B to G); the evaporation anomaly and absolute 10m sea wind (Figure 10-B to D). For a vertical cross-section of the troposphere along $30°N$ (cf. white horizontal line in Figure 2-A) we also investigate the anomaly of geopotential height, temperature, specific humidity and meridional wind (Figure 3). The section cuts across the Indus Plain between $70°$and $78°E$, just south of the UIB, where the southerly advection originates. In each Figure, the lead/lag is relative to the timesteps selected for the composite.

For a more quantitative approach, a PC regression is used, similar to the one developed in the previous section, but with different variables. The regression predicts $W700$ using 3-hourly time series of three different 2D fields: geopotential height at $300\,\mathrm{hPa}$, geopotential height thickness between $700$ and $500\,\mathrm{hPa}$, and between $500$ and $300\,\mathrm{hPa}$. The predictors are taken





within a box between $50°$ and $90°E$, and between $20°$ and $50°N$ at $1°$resolution [2] except for the lowest thickness where grid points whose surface is above $700\,\mathrm{hPa}$ are removed (cf. the extent of the colour shading shown in Figure 5-B, C, and D). Both

positive and negative values of $W700$ are predicted. The time series of the predictors are standardised (i.e. the mean is removed and the result divided by the standard deviation), and the PC decomposition is then performed independently for each 2D field. Finally, the regression makes use of the first 40 PCs of each 2D fields, for a total of 120 predictors.

     The regression has a high predictive skill, with $R^2 = 0.886$. The scatter plot of $W700$ against its prediction shown in Figure 5-A also suggests that the prediction is successful for a wide range of values of $W700$, despite some underestimations for

the highest values. The coefficients of the regression are displayed in Figure 5-B, C and D. The partial prediction of $W700$ associated with each of the three fields are called $Z300$ for the geopotential at $300\,\mathrm{hPa}$, and $dZ\_UP$ and $dZ\_LW$ for the geopotential thickness between the layers 500-300 hPa and 700-500 hPa respectively. Note that a more detailed explanation of the methodology is available in Baudouin et al. (2020a) and Baudouin (2020).

     The geopotential height at $300\,\mathrm{hPa}$ has been used for tracking analysis (Cannon et al., 2016) and WD indices (Madhura

et al., 2015; Midhuna et al., 2020) to characterise WDs. However, only using geopotential heights as predictors limits the predictive skill ($R^2 = 0.678$), and leads to an underestimation of $W700$ for values above $2.5\,\mathrm{m\,s^{-1}}$ (Baudouin, 2020). This issue is corrected by adding geopotential height thicknesses as predictors. That way, the regression can be interpreted as the decomposition of the wind at $700\,\mathrm{hPa}$ into the wind at $300\,\mathrm{hPa}$ and the windshear between the two layers.

     We noticed that $dZ\_UP$ and $dZ\_LW$ are negatively correlated with the predictant $W700$. In regression studies, predictors

exhibiting such a behaviour are referred to as a suppressor: the predictor suppresses the variability of one or several other predictors (Smith et al., 1992; Nathans et al., 2012). In this case, $dZ\_UP$ and $dZ\_LW$ suppress some of the variability of $Z300$. To simplify the analysis, a new contribution of the geopotential thickness to the wind is computed. $dZa\_UP$ is the residual of the regression of $dZ\_UP$ with $Z300$: the variability of $Z300$ is removed from $dZ\_UP$. It describes the anomalous geopotential thickness given the situation of the geopotential at $300\,\mathrm{hPa}$. The same is performed for $dZ\_LW$. The regression

is summarised in the equation below, with $\widehat{W700}$ as the prediction of $W700$:

$$
\begin{aligned}
\widehat{W700} &= \beta_0 + Z300 + dZ\_UP + dZ\_LW \\
&= \beta_0 + Z300 + (a \times Z300 + dZa\_UP) + (b \times Z300 + dZa\_LW) \\
&= \beta_0 + (1 + a + b) \times Z300 + dZa\_UP + dZa\_LW
\end{aligned}
\tag{1}
$$

     Note that $Z300$ is recomputed hereafter so that it includes the fractions a and b derived from $dZ\_UP$ and $dZ\_LW$, respectively. Note also the presence of the constant offset $\beta_0$. The same composites based on the highest value of $W700$ as above are used to investigate the evolution of those contributions during a peak of $W700$ (Figure 6).

---

[2]Higher resolution is not needed and requires significantly higher computational power.





### 3.3 Investigating variability in WD structure

There is variability in the dynamic structure of WD. To investigate this variance, a more complex composite analysis is used, based on a quantile regression. Two subsamples of the time steps with $W700$ above the $90^{th}$ percentile are created, so that they both have the same mean geopotential anomaly at $300\,hPa$, but one is related to lower values of $W700$ (although still above the $90^{th}$ percentile) and the other to higher values. Two quantile regressions are used, each predicting the first and the third quartile of $\widehat{W700}$ (the prediction of $W700$ with geopotential heights and thicknesses, cf. Equation 1), as a function of $Z300$, of geopotential anomaly at $300\,hPa$ at the centre of the mean WD (66°E - 36°N), and of months. The group with lower (higher) $W700$ is composed of all the time steps with $W700$ below (above) the first (third) quartile computed by the regressions. Since the ratio between $Z300$ and $W700$ is lower before the wind peak than after (cf. Figure 6), using only $Z300$ in the quantile regressions would lead to an over-representation of time steps with WDs closer to (further away from) the UIB in the low (high) wind group. Including the geopotential anomaly at 300 hPa at the centre of the mean WD removes this issue and insures that the mean WD intensity is similar in both groups. Finally, the months are introduced in the quantile regressions, so that the composites of the anomaly presented below are not affected by the seasonality: for each month, the same number of time steps is present in each group. Note that the quartiles of $\widehat{W700}$ are predicted and not that of $W700$ itself, so that the differences between the two groups are not related to the variability missed by the PC regression of $W700$ with geopotential heights and thicknesses. Figures 7 and 8 represent the same variables as in Figures 2 and 4 respectively but based on the new composites.

Finally, a second sampling is performed to investigate the specific humidity variability in WDs. Again, each sub-selection includes a quarter of the time steps with $W700$ above the $90^{th}$: one characterises low $Q700$, and the other high $Q700$. Therefore, the quantile regression predicts the first and the third quartile of $Q700$. The predictors used are the months, to remove the impact of seasonality, and the difference of the geopotential height anomaly between the grid points 60°E-36°N and 70°E-36°N. This latter predictor fixes the longitudinal gradient of geopotential across the UIB, and in that way the position of the mean WD, thereby avoiding over-representing timesteps before (after) the maximum of $W700$ in the group of high (low) $Q700$ (cf. Figure 6). However, the intensity of the WDs is not fixed as in the previous case. Various composites are derived from theses selections: Figures 11, and 12, respectively comparable to Figures 9 and 2.

## 4 Results on the wind contribution

### 4.1 WD characteristics

#### 4.1.1 The upper troposphere disturbance

High $W700$ is associated with a negative geopotential anomaly (i.e. cyclonic disturbances) with the minimum located at 36°N 66°E, just north of the Hindu Kush, and north-west of the UIB (Figure 2-B), and near the tropopause, at around $300\,hPa$ (Figure 3-B), in agreement with previous studies (Hunt et al., 2018a; Midhuna et al., 2020). The lead/lag analysis (Figures 2-A, 2-C,



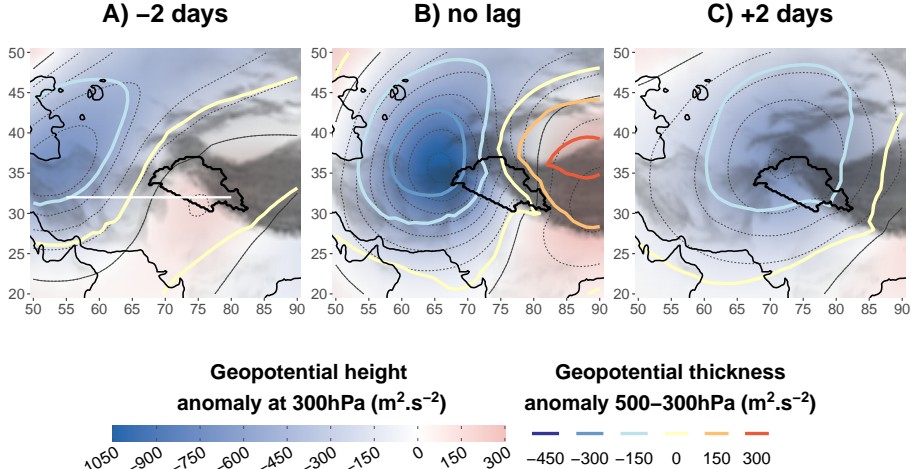

**Figure 2.** Composite maps of a geopotential height anomaly at $300\,\mathrm{hPa}$ (colour shading, thin contour lines every $150\,\mathrm{m^2\,s^{-2}}$) and a geopotential thickness anomaly 500-300 hPa (thick contour lines). Values are an average based on the 10% highest values of the $700\,\mathrm{hPa}$ wind contribution during winter ($W700$, panel B). For panel A (C) a two days lead (lag) is applied to the selection. The white line in panel A indicates the cross-section in Figure 3. Relief (grey shading) and coastline are based on ERA5 data. The UIB is indicated in each panel. Non-significant anomalies at the level 95% are shown in white (result of a t-test on the means). Note that the thickness anomaly is directly indicative of temperature anomalies in that layer under the hydrostatic equilibrium

3-A, and 3-C) reveals an eastward motion of the disturbance. Both characteristics fit the definition of Western Disturbances (WDs).

Interestingly, there is no evidence of a separate cyclonic circulation near the surface, neither on the day of high $W700$ (Figure 3-B), nor the day before (Figure 3-A), as mentioned by Dimri and Chevuturi (2014) and Dimri et al. (2015) among others. Instead, Figure 3-B shows that the lower altitude circulation is simply a weaker extension of the anomaly at $300\,\mathrm{hPa}$. This downward extension of the cyclonic circulation is also evident in the meridional wind field (Figure 3-D). However, Dimri and Chevuturi (2014) might have referred to fast-moving small-scale eddies that can circulate near the surface of the Indus Plain and are not detected by the composite analysis.

### 4.1.2 The cold core

The weakening of the cyclonic circulation towards the ground is due to a negative temperature anomaly: the WD has a cold core. The centre of the cold core is slightly displaced to the north-west of the geopotential anomaly minimum (Figure 2-B), which indicates baroclinicity. This baroclinicity is even clearer in Figures 3-A to C. There, a vertical black line indicates the location of the maximum anomaly for each altitude and broadly corresponds to the contour of zero meridional wind anomaly (Figure 3-D to F). The line exhibits a westward tilt with the altitude which is caused by the asymmetry between temperature and



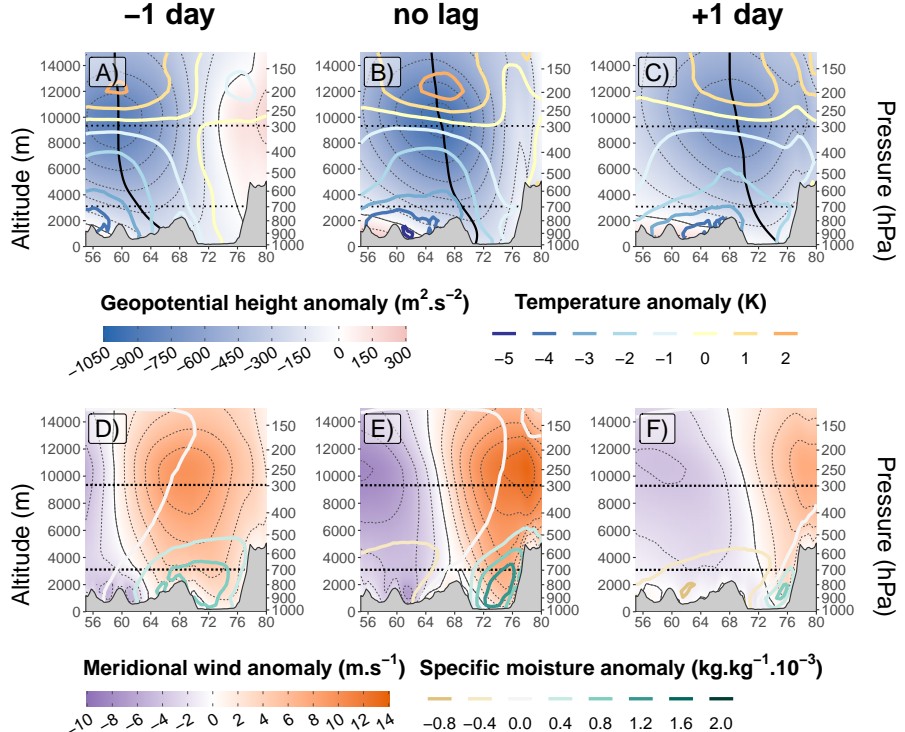

**Figure 3.** Composite cross-section along 30°N of anomaly of geopotential height (colour shading, thin contour lines every $150\,\mathrm{m^2\,s^{-2}}$) and temperature (thick contour line) in panels A to C, and meridional wind (colour shading, thin contour lines every $2\,\mathrm{m\,s^{-1}}$) and specific moisture (thick contour line) in panels D to F. Values are an average based on the 10% highest values of $W700$ (B, E). For panel A and D a one day lead is applied to the selection, while a one day lag is applied for panel C and F. The dotted horizontal lines represent the levels 700 and 300 hPa, the altitudes for the cross-barrier moisture transport in Figure 1 and minimum geopotential anomaly in Figure 2, respectively. The thick vertical black line in A, B, and C represents the longitude of the minimum geopotential anomaly as a function of altitude. The grey shading represents the relief as in ERA5. Note that the mean meridional wind is close to 0 except near the relief over the Tibetan and Iranian plateaus, where valleys funnel the mean westerly wind.

geopotential height, a characteristic of the baroclinic structure (Dimri and Chevuturi, 2014; Hunt et al., 2018a). The asymmetry
further increases at lower altitude, below $700\,\mathrm{hPa}$, where cold continental dry air is advected from Siberia and Central Asia at the rear of the WD (Figure 3, Yadav et al., 2012). The Sistan plain in eastern Iran forms a north-south valley between the Hindu Kush and the Iranian plateau, around 62°E in the cross-section, which funnels the cold air flow (Figure 3-B and E). Surprisingly, weaker but still negative temperature anomalies are also present in the Indus Plain, despite the southerly wind bringing in warmer air (Figure 3-B and E). This feature may be related to the adiabatic cooling (i.e. large scale upward
motions) resulting from the baroclinic instability. Closer to the surface, precipitation evaporation and reduced solar radiation could further explain the cooling.

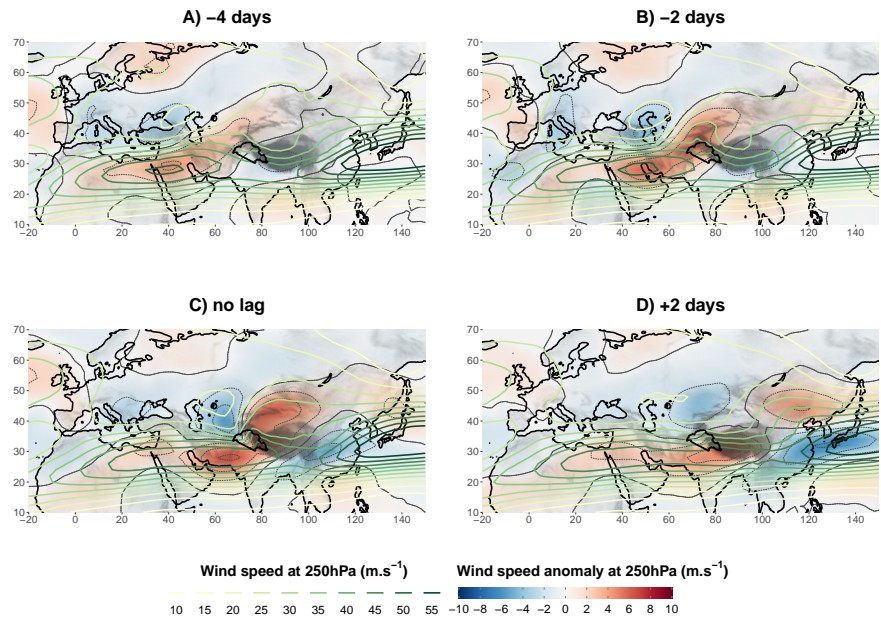

**Figure 4.** Composite maps of wind speed (thick contour lines) and wind speed anomaly (colour shading, thin contour lines every $2\,\mathrm{m\,s^{-1}}$) at $250\,\mathrm{hPa}$, based, for panel C on the same selection of the 10% highest $W700$ as in Figure 2. For panel A, B and C, a 4 day lead, a 2 day lead, and a 2 day lag, is respectively applied to the fields. Non-significant anomalies at the level 95% are shown in white.

### 4.1.3 Linking the WDs to the upper-tropospheric wind: jet-stream and outflow

WDs are embedded in the Subtropical Westerly Jet (SWJ Dimri et al., 2015). The SWJ is present in winter at around $250\,\mathrm{hPa}$ and between 25°N and 35°N (cf. Figure 4 Schiemann et al., 2009). It exhibits two local maxima independent of WDs: the strongest south of Japan (East Asian Jet, above $55\,\mathrm{m\,s^{-1}}$), the other between Egypt and Iran (Arabian Jet, above $45\,\mathrm{m\,s^{-1}}$).

The mean WD centre is located just north of the jet, and drives a north-south dipole of wind speed anomaly that progresses eastward, following the WD motion (Figures 2 and 4). The positive wind anomaly results in the narrowing and strengthening of the Arabian Jet. Dimri et al. (2015) suggest that part of the strengthening could be due to the merging of the SWJ with the polar jet, which marks the limit between cold and warm air at the surface. This merging is not evident from the figure 4, but the composite analysis may hide this dynamic due to the highly variable position of the polar jet. After the peak of $W700$, the two parts of the dipole dissociate themselves: the negative northern anomaly slowly moves north-eastwards, while the positive anomaly quickly continues eastward, south of the Tibetan Plateau, carried by the SWJ (Figure 4-D).

When the lag is negative (Figure 4-A and B), the positive anomaly of wind speed further extends to the north-east, towards Central Asia, outside of the jet core. This deformation corresponds to the second stable position of the SWJ, north of the Tibetan Plateau, which is predominant in summer, but may also occurs in winter (jet split, Schiemann et al., 2009). A picture from Pisharoty and Desai (1956), still in use in Dimri et al. (2015), suggests on the contrary that stronger wind speed occurs





at altitude at the rear of a WD. While some WDs may have this characteristic, WDs triggering high $W700$ and thus high precipitation do not exhibit that feature.

As the WD approaches the Tibetan Plateau, a second maximum positive anomaly of wind speed becomes evident over the high elevations: first over the Pamir range (Figure 4-B) and then along the Himalayas (Figure 4-C and D). This anomaly is in part the result of the increased funnelling of the jet over the high ground. At maximum $W700$, this positive anomaly extends to the north-east of the WD and corresponds to the outflow of the cross-barrier wind in the UIB. The outflow is characterised by a swift anticyclonic turn, with strong similarity to the warm conveyor belt associated with mature baroclinic waves (Martínez-Alvarado et al., 2014). The increase in latitude and the latent heat release are both key to explaining that change in relative vorticity (Grams et al., 2011). The outflow can also be revealed by the analysis of the cloud cover and the development of large bank of cirrus (Agnihotri and Singh, 1982; Rakesh et al., 2009; Hunt et al., 2018a).

Finally, after the peak of $W700$, the positive anomaly of wind speed related to the outflow completely splits from the circulation associated with the WD and continues eastward, north of the East Asian Jet. By contrast, the WD-relief interaction contributes to lower the intensity of the SWJ downwind: the East Asian Jet is notably weakened as a result (Figure 4-D). Meanwhile, the Arabian Jet remains anomalously strong; in fact, two days after the peak of $W700$ the strength of the anomaly is about the same as four days before the peak. Hence, the increased intensity of the Arabian Jet is a longer-term feature that seems to promote the intensity or occurrence of a WD, but is not directly affected by the passing of one. This feature is evident from several intra-seasonal and inter-annual studies (Filippi et al., 2014; Hunt et al., 2018a; Ahmed et al., 2019). Hunt et al. (2018a) also discussed the influence of the SWJ position on WDs, but the synoptic analysis presented here does not indicate that WDs are related to a change in the position of the Arabian jet.

### 4.2 Explaining $W700$ variability using the PC regression

### 4.2.1 Interpretation of the PC regression

The results of the PC regression developed in Section 3.2 are presented using Figure 5. Two patterns are distinguishable in the coefficients for $Z300$ (Figure 5-B). The first is a dipole across the UIB, indicative of a geopotential gradient, which translates into a south-westerly geostrophic wind for positive $W700$. The second, less clear, is a ring of positive coefficient around the negative centre of the dipole. The related gradient indicates a cyclonic anomaly for positive $W700$, and therefore a WD. The extent of the ring can be interpreted as the effective radius of the centre of the anomaly in triggering a cross-barrier wind in the UIB. The centre of the pattern is located to the west of the study area, where the relief forms a notch important to trigger the vertical velocities (cf. Figure 1 in Baudouin et al., 2020a; see also Lang and Barros, 2004; Cannon et al., 2015). In conclusion, $Z300$ is representative of a cyclonic south-westerly geostrophic wind at $300\,\mathrm{hPa}$.

The coefficients associated with $dZ\_UP$ and $dZ\_LW$ are very similar and represent a dipole at a similar location as the one discussed for $Z300$ (Figure 5-C and D). The related gradient is equivalent to a north-easterly thermal wind, that is, an increase in south-westerly wind towards lower altitudes. However, the composite analyses showed that the cyclonic circulation associated with WDs weakens towards lower altitudes (Figure 3-B), and that the southerly wind is weaker closer to the surface (Figure



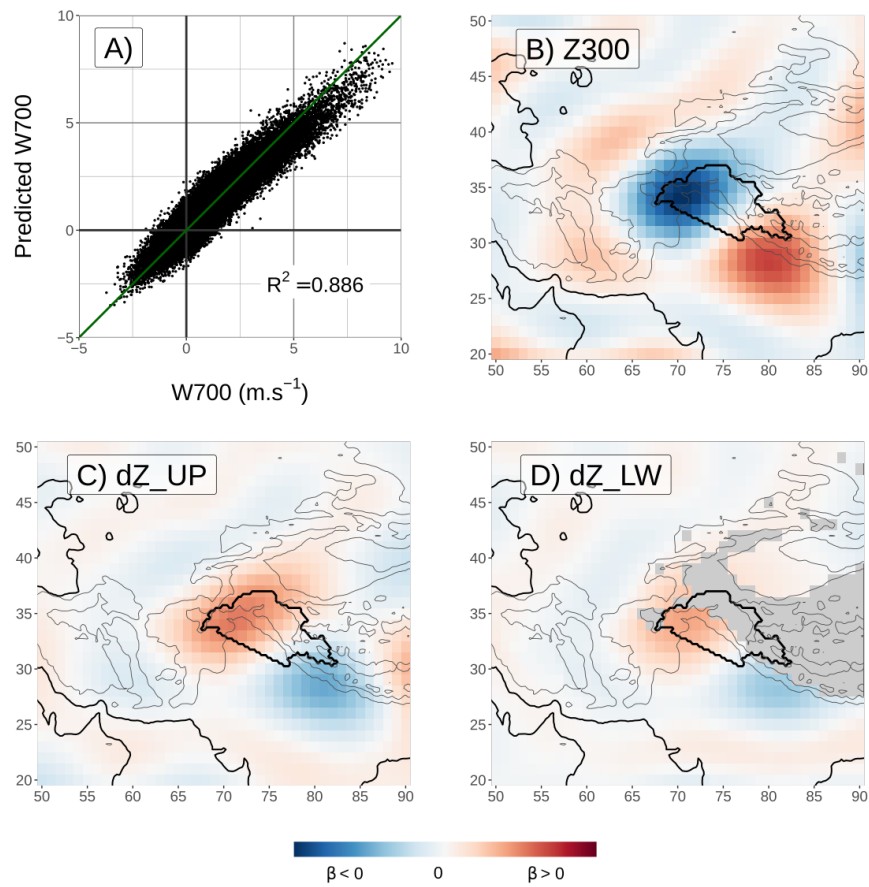

**Figure 5.** Results of the regression of 3-hourly $W700$ with geopotential height at $300\,\mathrm{hPa}$ ($Z300$), geopotential thickness 500-300 hPa ($dZ\_UP$), and geopotential thickness 700-500 hPa ($dZ\_LW$). Panel A is the scatterplot between the predictand and the prediction. Panels B, C, and D show the coefficients associated with each predictor (cf. Baudouin et al., 2020a, equation 5). The exact values of the coefficients are not interpretable and are therefore not indicated.

3-E). Indeed, $dZ\_UP$ and $dZ\_LW$ are negatively correlated with $Z300$. The new contributions $dZa\_UP$ and $dZa\_LW$ are designed so that they do not correlate with $Z300$ (cf. Section 3.2), rather they characterise the thermal structure of the WDs. If $dZa\_UP$ or $dZa\_DW$ are positive (negative) then the northwest-southeast temperature gradient over the UIB is weaker (stronger) than usual, and, consequently, the cold core of the WD is smoother, weaker, or further away (deeper or closer). These characteristics of the cold core are also evident from the composite maps for different $W700$ intensity (Figure 7-B and E). The importance of the thermal structure has been little investigated previously. Midhuna et al. (2020) used geopotential thickness to characterise WD intensity. They suggested that a colder WD (a thinner 850-200 hPa layer) characterises a more intense WD. The analysis here, however, shows on the contrary that a WD with a warmer core than usual (or displaced cold core) produces stronger near surface dynamics that enhance precipitation.





Interestingly, the patterns for all predictors (Figure 5-B, C, and D) suggest that $W700$ is sensitive to a geostrophic wind
that is rotated by approximately $45°$ compared to the southerly wind that it is meant to predict (cf. Figure 1). Although the
geopotential gradient is sufficient to predict wind intensity, it does not indicate the wind direction, which bends towards the
lower pressures, as noted in Baudouin et al. (2020a). In this paper, it is suggested that this ageostrophic effect is caused by the
drag along the Himalayan barrier and that it is key to trapping the cross-barrier flow in the notch formed at the intersection
between the Himalayas and the Hind Kush.

Finally, the regression successfully predicts the negative values of $W700$ (Figure 5-A). These correspond to northerly di-
vergent winds at $700\,\mathrm{hPa}$ in the UIB. The regression suggests that they are related to anticyclonic north-easterly geostrophic
winds. Since these winds are much less common than their south-westerly counterparts, large negative values of $W700$ are
rarer than positive ones. The regression also slightly underestimates the highest values of $W700$, possibly because the larger
latent heat release and increased buoyancy in the context of stronger convergence further sustains cross-barrier winds before
the pressure gradient can adjust to it.

#### 4.2.2 Evolution of the partial predictions

The different partial predictions of $W700$ vary during a wind peak event as it is evident in Figure 6. The peak value of $W700$
(i.e. at no lag) is first due to $dZa\_UP$ (accounting for 46% of the predicted value), highlighting the importance of considering
the WD thermal structure. $Z300$ is close behind (43%), while $dZa\_LW$ has a minor role (11%). Interestingly, $W700$ (the
dashed orange line) raises and decays symmetrically around its peak. However, this is not the case for its different partial
predictions (other dashed lines). All partial predictions rise steadily before the wind peak. However, the two partial predictions
associated with geopotential thicknesses peak before $W700$ reaches its maximum intensity and quickly drop after: $W700$ is
only maintained by $Z300$ after its peak. The asymmetry between $Z300$ and $dZa\_UP$ can be investigated with Figures 2 and 3.
Focusing on Figure 2-A, two days before the wind peak, a gradient in geopotential is already present over the UIB in relation
to the positive geopotential anomaly ahead of the WD (increasing $Z300$). This gradient triggers a southerly advection and a
warm anomaly that limits the presence of any gradient in temperature (increasing $dZa\_UP$). At the wind peak (Figure 2-B),
the approaching WD tightens the gradient of the $300\,\mathrm{hPa}$ geopotential while the colder air remains at the edge of the IUB (peak
of both $Z300$ and $dZa\_UP$ contributions). Two days after the wind peak at $700\,\mathrm{hPa}$ (Figure 2-C), the WD enters the UIB with
its cold core. The geopotential gradient remains tight, but the increased temperature gradient counterbalances it, so that the
southerly wind at $700\,\mathrm{hPa}$ weaken ($Z300$ remains high but $dZa\_UP$ quickly decrease).

### 4.3 Geostrophic processes and baroclinic interaction

#### 4.3.1 Importance of the upper-troposphere geopotential gradient

The PC regression indicates that the presence of an east-west geopotential gradient across the UIB is more important than
the proximity of a WD (Figure 5-B). The importance of this gradient is confirmed by the presence of a positive geopotential



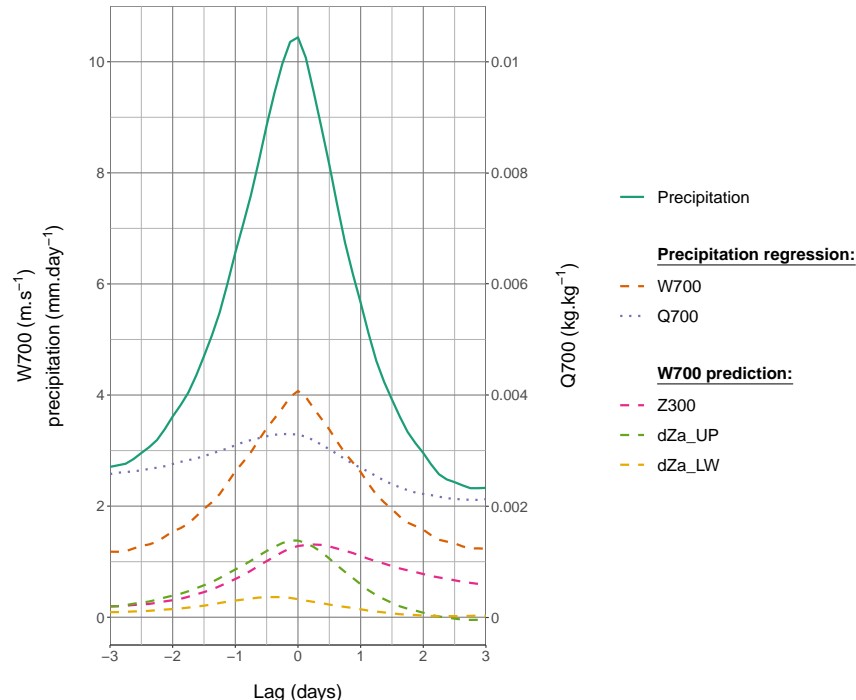

**Figure 6.** Lead/lag composite around the 10% highest values of $W700$, as in Figures 2 3 and 4, for ERA5 precipitation, contributions to precipitation based on moisture transport ($W700$, $Q700$, cf. Section 3.1), and partial prediction of $W700$ based on geopotential ($Z300$) and geopotential thicknesses anomaly ($dZa\_UP$ and $dZa\_LW$, cf. Section 3.2). Note that $Z300$, $dZa\_UP$, and $dZa\_LW$ have the same units as $W700$, and, with the intercept, add up to the prediction of $W700$ ($\widehat{W700}$, cf. Equation 1).

anomaly over the Tibetan Plateau during high $W700$ events (Figure 2-B). The intensity of this anomaly is also clearly coupled with that of $W700$ (Figure 7-B and E).

     This anticyclonic anomaly is already present over the UIB two days before the wind peak (Figure 2-A) and in conjunction with the WD forms a wave train that is already discussed in the literature (Dimri, 2013; Hunt et al., 2018b). However, these previous studies have not stressed the importance of this anticyclonic extreme, which increases the zonal gradient of geopo-

tential to the east of the WD and thus strengthens the southerly wind. The downward propagation of that gradient of pressure is sufficient to trigger the cross-barrier meridional flow. In fact, the zonal gradient is already present at $700\,\mathrm{hPa}$ one day before the wind peak, despite the mean WD centre located at about $1500\,\mathrm{km}$ to the west, owing to the tilt of the circulation and the presence of positive geopotential anomalies towards the Himalayas. Hence, cross-barrier wind and precipitation can all occur well ahead of a WD. By contrast, when the WD is above the UIB, the zonal gradient of geopotential at $700\,\mathrm{hPa}$ disappears,

stopping the cross-barrier meridional flow (Figure 3-C and F).

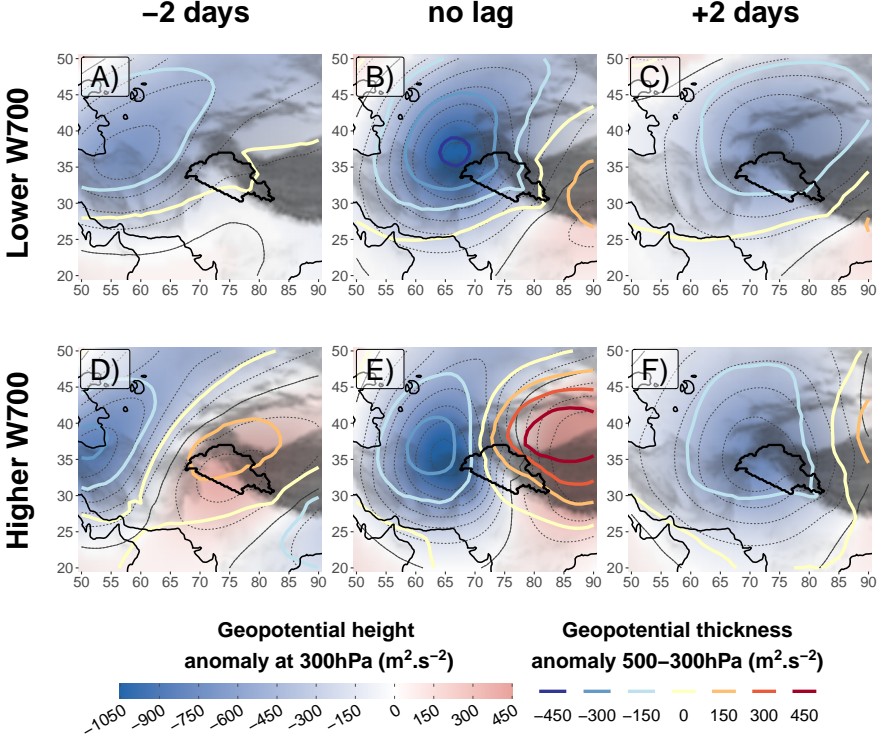

**Figure 7.** Same as Figure 2, but the composites are based on a subsampling of lower (higher) values of $W700$, while still above the $90^{th}$ percentile, in panels A to C (D to F). The subsampling is made so that it is not impacted by seasonality nor by the intensity of the WD (cf. Section 3.3. This way, comparisons can be made with Figure 2.

### 4.3.2 Changes in WD motion speed

The speed of the mean WD can be calculated using the lead/lag analysis: the centre moves by $13°$ of longitude in two days between Figure 2-A and Figure 2-B, indicating a mean speed of about $6.8 \, \mathrm{m \, s^{-1}}$, which is not far from the phase velocity of $5 \, \mathrm{m \, s^{-1}}$ found in Hunt et al. (2018b), but is slower than the 8 to $10°$ per day found in Datta and Gupta (1967) (cf. Dimri et al.,

2015). Interestingly, Figure 2-C shows a slowing of the mean WD to about $3.1 \, \mathrm{m \, s^{-1}}$, although the composite method limits the analysis in terms of mean speed of the WDs. Slower or even static cyclonic circulation on the surface is sometimes discussed in the literature (e.g. Lang and Barros, 2004; Dimri and Chevuturi, 2014), but not the slowing of the tropopause disturbance itself.

The speed of the WD also varies depending on the peak intensity of $W700$ (Figure 7). Before the peak of $W700$, the WD is

325 moving significantly faster in the group with the highest peak values of $W700$ ($6.8 \, \mathrm{m \, s^{-1}}$, Figure 7-D and E) than in the other group ($4.2 \, \mathrm{m \, s^{-1}}$, Figure 7-A and B). This change in translation speed does not seem to be related to a change in the $250 \, \mathrm{hPa}$ wind speed (Figure 8). We rather suggest it to be related to a difference in the mechanisms explaining the development of the wave supporting the WD.



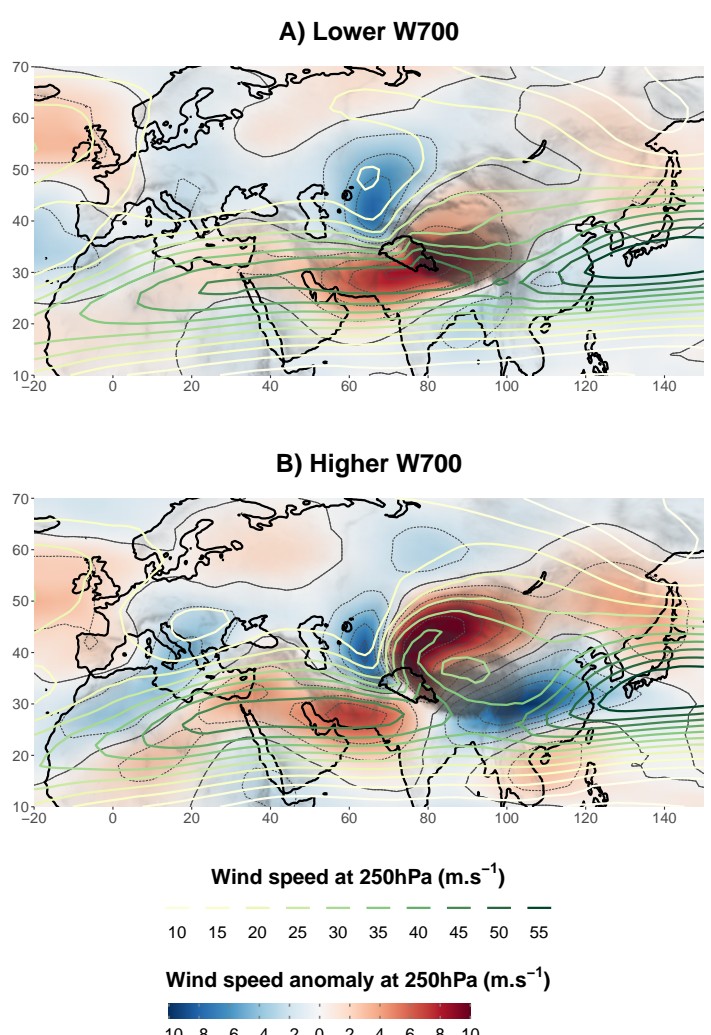

**Figure 8.** Same fields as in Figure 4-C, but for composites based on the same selections as in Figure 7-B and E.

In the lower $W700$ case, the development of the WD is accompanied by the reinforcement of the cold anomaly (Figure
7-A and B), suggesting that the tropopause anomaly mostly grows from the equatorward motion of cold air at the rear of the
WD. This growing process results in a steady upwind propagation of the wave. By contrast, the temperature anomaly does not
change as the WD with higher $W700$ approaches the UIB and strengthens (Figure 7-D and E). In this case, the WD doesn't
grow, but propagates faster.

Furthermore, a stronger cross-barrier wind results in a more severe slowing of the WD. After the peak of $W700$, the WD
with higher $W700$ slows to $3.2\,\mathrm{m\,s^{-1}}$ (Figure 7-E and F), even slower than the WD with lower $W700$ ($3.8\,\mathrm{m\,s^{-1}}$, Figure 7-B
and C). The intensifying uplift in the UIB during a high cross-barrier wind event increases the near tropopause convergence





over the UIB (cf. Quadrupole on divergence field in Hunt et al. 2018a). This convergence zone results in the decay of the WD from its east, and therefore the WD's slowing. This process corresponds to the last and decaying stage of a baroclinic interaction which is now further investigated.

### 4.3.3 Thermal structure feedback

A first simple way to describe the baroclinic interaction at play in the UIB is through the feedback between circulation and thermal structure. The analysis of the PC regression reveals the importance of the horizontal temperature gradient at $300\,\mathrm{hPa}$ and how a displaced or weaker cold core enhances the circulation at $700\,\mathrm{hPa}$. Yet, the southerly wind east of the WD brings warm air to the Indus plain at all altitudes (Figure 3), depending on the large scale north-south temperature gradient. This flow therefore contributes to displacing the cold core of the WD to the west of its circulation centre at the tropopause, and to reinforcing the $700\,\mathrm{hPa}$ circulation in the Indus. The Iranian plateau and the Hindu Kush act as a barrier to the arrival of the cold air from the north-west, while the flat Indus plain channels the southerly wind, further enhancing the positive feedback process.

At local scale, the UIB is a hotspot where most of the near surface convergence occurs. There, warm advection, latent heat release, and mixing prevents the cold air from extending into the UIB as the WD progresses eastwards. The zonal temperature gradient at $300\,\mathrm{hPa}$ is therefore reduced, which in turn increases the geopotential height gradient at $700\,\mathrm{hPa}$ and thus the cross-barrier wind and the uplift (cf. analysis of the PC regression). This effect explains the general potential for intense cross-barrier winds in the area. The cold core eventually enters the UIB when the cold air invades the Indus plains to the the South of the UIB and stops the advection of warm air.

### 4.3.4 Ageostrophic circulation induced by baorclinicity

Baroclinic interaction can also be described using quasi-geostrophic theory and potential vorticity (PV). When a positive PV anomaly evolves in a sheared environment, baroclinically forced vertical velocities occur: uplift upwind of the anomaly, and subsidence downwind (Malardel, 2005). More specifically, it is the zonal gradient of PV ahead of (behind) the PV anomaly that triggers the baroclinic uplift (subsidence). A baroclinic interaction occurs when a surface and a tropopause anomaly align such that the vertical velocities associated with one enhances the other anomaly by stretching it. The two disturbances grow and their associated vertical velocities increase until the shear leads to the spatial dephasing of the disturbances. Then, a reverse negative feedback process takes place, leading to the quick decay of both disturbances. These processes characterise the most intense cyclonic activity (Malardel, 2005; Dacre et al., 2012)

A WD is characterised by a high PV anomaly (Hunt et al., 2018a), and the SWJ in which it is embedded provides the shear needed to produce the vertical velocities. This process has already been suggested as a key factor for explaining vertical velocities in the UIB (Sankar and Babu, 2020) or the interaction with tropical depressions (Hunt et al., 2021). However, as discussed earlier (Section 4.1.1), there is no evidence of a near surface disturbance that would trigger a baroclinic interaction. This has lead Hunt et al. (2018a) to describe WDs as "immature baroclinic waves". However, we suggest that the relief and the orographic uplift provides the low altitude feedbacks for a similar, yet short lived, baroclinic interaction.




The zonal gradient of PV upwind of the WD triggers a baroclinic uplift to the east of the anomaly. As the WD approaches the UIB, its interaction with the relief results in an orographic uplift, which combines with the baroclinic uplift. However, the uplift also results in decreased PV at high altitude (300 hPa) over the UIB due to the advection of lower tropospheric PV as well as diabatic heating. Hence, the zonal PV gradient increases and stronger baroclinic uplift occurs, resulting in a positive feedback. Once the high PV anomaly associated with the WD enters the UIB, the mechanism reverses: the orographic uplift results in the weakening of the PV anomaly, eventually leading to the decay of the WD.

The input of low PV at high altitude from the UIB is evident from the anticyclonic turn of the outflow (cf. discussion on Figure 4-C, in section 4.1.3) and from the warm area at 300 hPa to the east of the WD but largely displaced compared to the geopotential anomaly maximum (Figure 2-B). Moreover, the baroclinic interaction and the intense uplift is revealed by the deformation of the jet near the UIB during the passing of the WD (Figure 4). Particularly, when the uplift is maximum, the jet configuration is characteristic of a strong divergence at the tropopause over the UIB: the study area is located near the eastern exit of the Arabian Jet, and in the right entrance of the outflow[3]. This mechanism is similar to the baroclinic interaction between a tropopause anomaly and a surface disturbance, but where the surface disturbance is replaced by the relief. It also results in a weaker, shorter-lived interaction.

The different thermal structures shown in Figure 7 are linked to a difference in the strength of baroclinic processes. This difference becomes evident when looking at the wind at 250 hPa. In case of higher W700 (Figure 8-B), the pattern of wind anomaly is very similar to Figure 4-C. The outflow is more intense and clearly detached from the anomalously strong SWJ. By contrast, in case of lower $W700$ (Figure 8-A), the anomalously strong SWJ extents to the South-East of the UIB and forms a diffluence over the Tibetan Plateau, while there is no evidence of a distinct outflow. In that context, the UIB is not any more located in a tropopause divergence configuration.

To conclude, WDs are a prerequisite to cross-barrier barrier winds in the UIB. Both the strength of the 300 hPa low and its baroclinic context are important for the cross-barrier wind intensity. In the composites in Figure 7, $W700$ is 2/3 higher in the higher case ($5.3\,\mathrm{m\,s^{-1}}$) than in the lower case ($3.2\,\mathrm{m\,s^{-1}}$), a difference mostly explained by the difference in baroclinicity. However, the increase in precipitation is almost 3 fold, from 5.9 mm·3hrs$^{-1}$ to 16.8 mm·3hrs$^{-1}$. This difference is partially explained by the quadratic fit (cf. Figure 1-B), but also by a change in moisture supply ($Q700$), from $3\times10^{-3}\,\mathrm{kg\,kg^{-1}}$ to $3.6\times10^{-3}\,\mathrm{kg\,kg^{-1}}$, which is now further investigated.

## 5 Relating moisture contribution variability to Western Disturbances

### 5.1 General

The UIB and the Indus Plain are dominated by a mean westerly moisture flux during the season of active WDs (Figure 9-A). However, precipitable water is over 15mm in the Indus Plain, while it is below 10 mm to the west of the plain, in relation to the higher elevations (e.g. Suleiman range, Hindu Kush, Iranian plateau). This being the case, transient circulation or local

[3]On a jet streak, the right entrance and the left exit are characterised by baroclinic convergence at the tropopause and tropospheric uplift (Malardel, 2005)




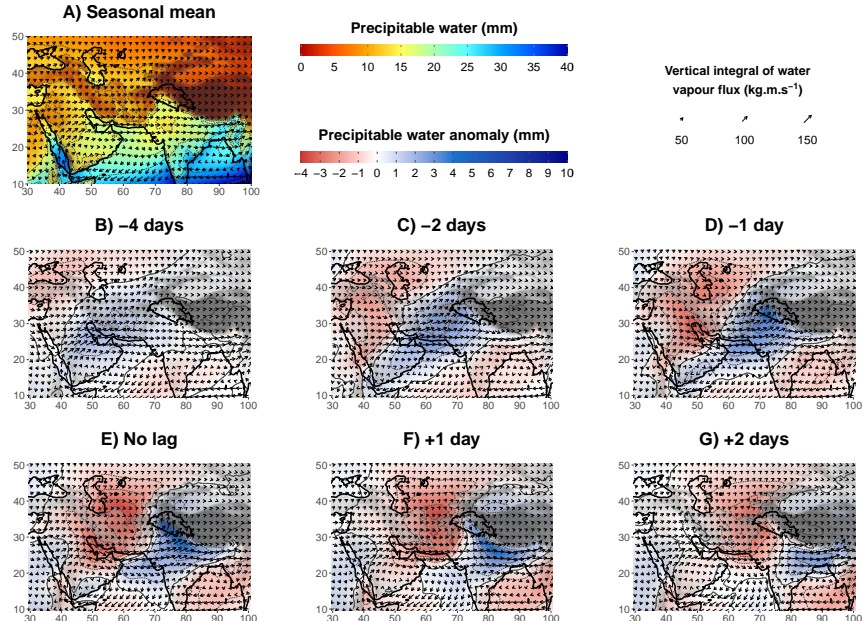

**Figure 9.** In Panel A, Seasonal mean of precipitable water (colour shading, thin contour lines every 5mm) and vertical integral of water vapour flux (arrows). The mean is weighted by the seasonality of occurrence of $W700$ above its $90^{th}$ percentile. Panels B to G show the lead/lag composite maps, with respect to the 10% highest values of $W700$ (as in Figure 2), of precipitable water anomaly (colour shading, thin contour lines every 2 mm) and absolute water vapour flux (arrow). Non-significant anomalies at the level 95% are shown in white.

evaporation are needed to explain the higher amount in the plain. Furthermore, the moisture content (and moisture contribution, $Q700$) rises during a peak of $W700$ and in the days before as shown in Figures 3-D and E, and 6, indicating the importance of WDs in moisture variability. Therefore, to understand the origin, pathway and variability of the moisture that eventually precipitates, the moisture flux needs to be investigated during the passing of a WD.

## 5.2 Moisture pathway to the UIB

Figure 9-E shows a significant accumulation of precipitable water in the UIB when $W700$ is maximum: the anomaly represents an increase of 30 to 50% in precipitable water in the UIB compared to the mean. The anomaly extends along the Himalayas, towards northeast India, as well as towards the Arabian Sea. Hunt et al. (2018b) found a similar pattern when investigating extreme precipitation events. This relationship between wind and precipitable water is first explained by the near surface
moisture convergence induced by the cross-barrier wind, particularly near the foothills. However, composites of negative lags in Figure 9-B, C, and D suggest that anomalously wet air masses advected into the UIB are also responsible for the increase in precipitable water.





Four days before the peak of $W700$, moisture is already building up over the Persian Gulf (Figure 9-B). The direction of moisture transport indicates that this moisture is advected from the Red Sea across the Arabian Peninsula. This transport, and
how it is enhanced by Mediterranean lows, is described in Chakraborty et al. (2006) and Mujumdar (2006). This moisture flow passes over the coastal mountain ranges along the Red Sea (Sarawat Mountains, with peak heights ranging from 1000 to 3000 m). The moisture is therefore present at a relatively high altitude, similar to that of the cross-barrier wind in the UIB (700 hPa). The more intense wind at this altitude than closer to the surface has the potential to quickly transport the moisture towards the study area.

Two days later (Figure 9-D), the positive anomaly of precipitable water has spread from the north side of the Arabian Sea to the Indus Plain and the mountains to the west. The south-westerly moisture flux clearly suggests that air masses from the north of the Arabian Sea and the Persian Gulf contribute to the increase in precipitable water, as mentioned by Hunt et al. (2018b). Moisture convergence over the windward side of the mountains also helps to increase moisture at higher altitudes as can be seen over the Suleiman Range, one day before the peak in Figure 3-D. Convergence also starts over the UIB, more specifically
in the notch formed by the mountain ranges, one day before the peak of $W700$, resulting in a maxima of precipitable water anomaly (Figure 9-D).

This analysis shows the existence of a pathway of moisture that sustains moisture content in the UIB. It originates from the Red Sea, crosses the Arabian Peninsula towards the Persian Gulf, continues towards the north of the Arabian Sea, the Indus Plain and ends in the UIB. Previous studies have implied the existence of such a pathway without investigating it further
(Filippi et al., 2014; Hunt et al., 2018b; Ahmed et al., 2019). Part of this circulation, from the Red Sea to the Arabian Sea, is visible in the seasonal mean (Figure 9-A) and is driven by the subtropical gyre located on the southeastern tip of the Arabian Peninsula. WDs strengthen this flow as suggested in Hunt and Dimri (2021) but most importantly steers it towards the UIB. In some extreme cases, this moisture pathway can form atmospheric rivers (cf. Bao et al., 2006; Zhu and Newell, 1998) that are related to extreme precipitation events along the Himalayas (Thapa et al., 2018). This analysis also suggests that neither the
Mediterranean Sea nor the Caspian Sea are important contributors of moisture for the precipitation in the UIB, which has been suggested by other studies (e.g. Dimri et al., 2015).

## 5.3 Evaporation sources

The local evaporation in the UIB is quite high compared to the surrounding land, about 1 to 2 mm per day (Figure 10-A) and is little affected by the passing of a WD (Figure 10-C). While this process helps sustain the generally higher precipitable water
in the IUB, it cannot explain the precipitable water anomaly associated with a WD. The rest of the Indus Plain, or the nearby mountains, contribute very little to the moisture, as those areas are arid, and the wetter and higher ground is too cold in winter to generate significant evaporation.

The mean evaporation rate over the Persian Gulf and the north Arabian Sea is about 4 to 5 mm per day (Figure 9-A). That is, a day of evaporation represents the quarter of the total precipitable water in the area. This high replacement rate suggests that a
large amount of the moisture arises from those water bodies directly, and it also emphasizes the importance of the persistence of the moisture pathway between the two to build up moisture content. Even stronger evaporation rates are present in the Red



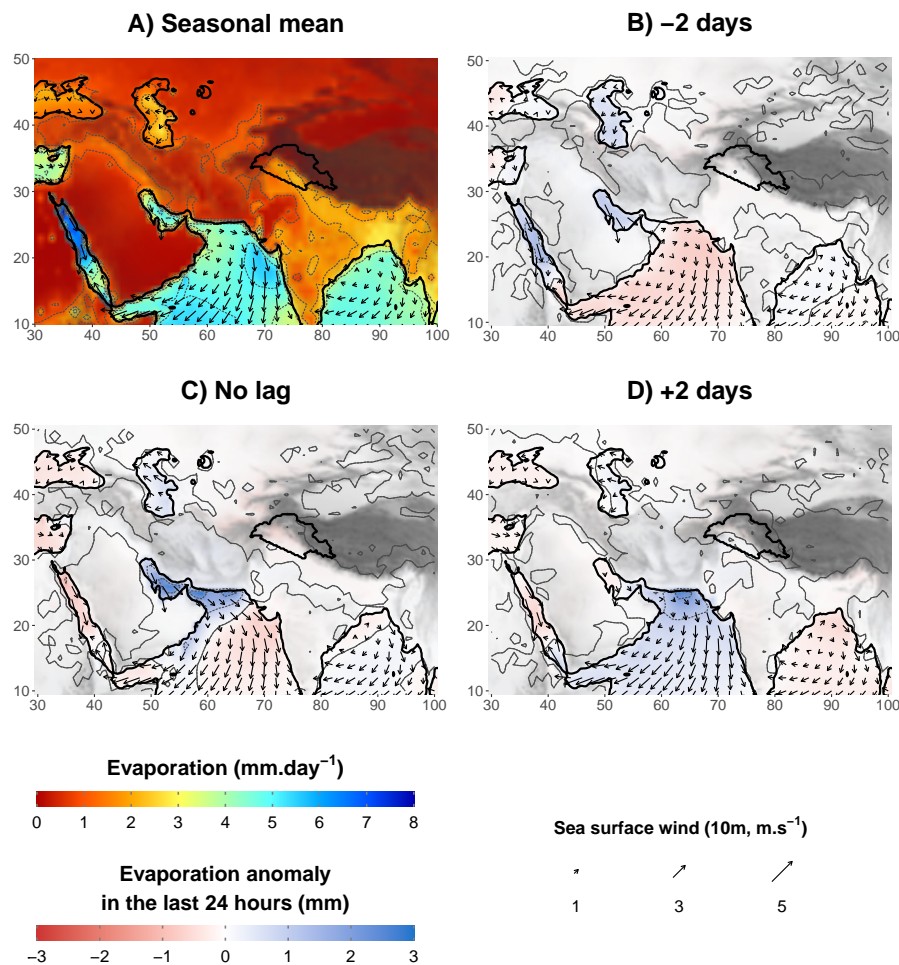

**Figure 10.** Panel A is similar to Figure 9-A but for evaporation (colour shading, thin contour lines every 5 mm) and 10m sea winds (arrows). Panels B to D are similar to Figure 9-C, E and G but for cumulated evaporation in the last 24 hours before the peak of $W700$ plus the lead or lag indicated (colour shading, thin contour lines every 2 mm) and 10m sea winds (arrows).

Sea, particularly to the north. As mentioned earlier, despite the distance, the Red Sea seams important for increasing moisture at higher altitudes, so that it can be more easily transported.

These two composite analyses clearly identify the north of the Arabian Sea, the Persian Gulf, and the Red Sea as the
main sources of moisture for the precipitation in the UIB for the first time. However, this approach does not allow for any quantification, nor does it discuss variability. A more complex analysis involving numerical tracking would be needed, and that is beyond the scope of this study.



### 5.4 Other impacts of the passing of WDs on the moisture field

WDs impact the intensity and end point of the moisture pathway (Section 5.2), but other impacts are evident from Figure 9 and
10 as well. Two days before the peak of $W700$, a negative anomaly of precipitable water propagates from Tigress-Euphrates
plain into the Persian Gulf (Figure 9-C), and reduces the moisture supply from the Red Sea. This circulation is the result of
a surge of a surface, cold, and dry air at the rear of WDs. The negative anomaly eventually propagates to the north of the
Arabian Sea after the peak of $W700$ (Figure 9-E, F, and G), but tends to weaken. This weakening can be explained by the
large evaporation anomaly present over the seas as the dry air progresses (Figure 10-B, C, and D). Evaporation rates almost
double compared to the seasonal mean (Figure 10-A), which is enough to eliminate the precipitable water anomaly in a day.
This evaporation anomaly can be explained by both the increase in surface wind speed induced by the WD (arrows on Figure
10) and the very low surface dew point of the air blown from the land.

A second but connected area of negative anomaly of precipitable water is present over Central Asia and the Caspian Sea
(Figure 9-C and D). It also relates to cold and dry air being advected by the WD from northern latitudes. When $W700$ peaks,
this cold and dry air mass invades the mountains to the west of the Indus Plain (Figure 9-E, see also Figure 3-B and D). Unlike
the other air masses that are advected over the sea, the encounter with the relief further increases the moisture anomaly through
a Foehn effect. This effect is strongest after the peak of $W700$, as the mean moisture flux over the Indus Plain veers eastward.
The continental dry air descends from the Suleiman Range and the Hindu Kush, replacing the maritime moist air (Figure 9-F
and G). As the UIB is cut from its moisture supply, the precipitable water anomaly becomes negative (Figure 9-G), which
severely limits moisture transport and precipitation despite the continued presence of a cross-barrier wind. Similarly, Figure 6
indicates that the passing of a WD noticeably reduces $Q700$ several days after the peak of $W700$.

Hence, a WD has two opposite effects on moisture content in the UIB: moistening before its passing, and drying afterwards.
Therefore, another ensuing WD may be inhibited, in terms of precipitation, by the dryer conditions dominating in the Indus
Plain and the UIB, suggesting a negative feedback effect. However, one could also imagine a positive feedback, where a quick
succession of WDs provides a continuous moisture supply from the maritime sources by inhibiting the dry air invasion. The
interaction between WDs is not further investigated here and is likely subject to high variability and dependent on larger and
smaller-scale atmospheric circulation.

After the peak of $W700$, the positive anomaly of precipitable water moves along the Himalayas towards northeast India
where it enhances cross-barrier moisture transport and orographic precipitation in another notch that is formed by the relief,
enhancing orographic precipitation there. The impact of WDs is not as evident in this area, first because the tropopause distur-
bances have mostly disappeared due to the interaction with the relief in the UIB, and because convection is a more important
driver of precipitation (Tinmaker and Ali, 2012; Mahanta et al., 2013; Mannan et al., 2017).

Finally, the north-easterly trade winds that blow over most of the Arabian Sea are also affected by WDs: they are weaker
ahead of the WD, leading to a decreased evaporation (Figure 10-B), which also slightly decreases precipitable water along
10°N (Figure 9-B to E). However, trade winds intensify after the passing of the WD, thus increasing evaporation (Figure 10-D)
and leading to a slight increase in precipitable water to the south (Figure 9-F and G).



## 5.5 Impact of WD characteristics on $Q700$

Various WDs characteristics influence moisture contribution ($Q700$). There exists a direct link between $W700$ and $Q700$: their peak is concomitant as a WD passes (Figure 6), and their correlation reaches 0.45 for the entire winter season (once the seasonal
mean is removed from $Q700$). This relationship is explained by the advection of moisture by the cross-barrier wind from both the south (from the moisture pathway) and the near surface (due to uplifting). Since the intensity and the thermal structure of a WD at $300\,\mathrm{hPa}$ impact $W700$ (cf. Section 4.2), they also impact $Q700$. However, the shape and direction of a WD has an even greater impact by acting on the balance between the supply of maritime moist air and the intrusion of continental dry air discussed earlier (Section 5.4). This impact is investigated using the results of the second quantile regression described in
Section 3.3, and composite maps over two selections, one representative of low $Q700$ and the other of high $Q700$ (Figures 11 and 12).

As expected, the two selections exhibit very different patterns of precipitable water anomaly when $W700$ is maximum (Figure 11-E and F). In case of high $Q700$, a high positive anomaly is present from the Arabian Sea to northeast India, with a maximum over the UIB, indicating a sustained supply of moisture. By contrast, a negative anomaly is present over the Indus
Plain in the case of low $Q700$, showing that continental dry air is already invading the plain, cutting off any potential moisture supply from the moisture pathway. A small positive anomaly remains in the UIB, probably thanks to the uplifting. Interestingly, no evident difference in moisture flux direction is visible between the two panels (Figure 11-E and F).

The explanation of the differences in moisture content lies in the circulation history. In case of high $Q700$, moisture over the Persian Gulf increase more than in the average case four days before the peak of $W700$ (Figures 9-B and 11-B), due to stronger
moisture transport across the Arabian Peninsula. This moisture anomaly moves towards the Indus Plain, the mountain ranges to the west, and even towards parts of Central Asia two days later (Figure 11-D), pushed by a strong southerly component of the moisture flux. The larger extent of the precipitable water anomaly than in the average case (Figure 9-C) effectively delays the intrusion of continental dry air (Figures 11-F and 9-E). In the low $Q700$ case, no such positive anomaly is present (Figure 11-A and C): the WD is not accompanied by a strengthening of the moisture pathway, and the southerly component of the moisture
flux is almost absent. Instead, negative anomalies of precipitable water build up over Tigress-Euphrates plain and Central Asia (Figure 11-C) and move eastward and south-eastward respectively towards the Indus Plain (Figure 11-C). These patterns show that the balance between moisture advection ahead of a WD and dry air intrusion at the rear may vary significantly.

This difference in moisture flux can be explained by the origins and tracks of WDs. In case of high Q700, the mean low at $300\,\mathrm{hPa}$ exhibits an eastward motion around 33-34° N, which is slightly more to the south than the general case (36° N). The
WD also exhibits a weaker cold core, and a larger zonal gradient of geopotential to its east than in the general case (Figures 2-B and 12-D), implying a stronger southerly wind at $700\,\mathrm{hPa}$. Both southern location and stronger southerly advection at $700\,\mathrm{hPa}$ explains how these WDs are able to draw more moisture from the sub-tropical latitudes along their path. Furthermore, stronger mid-tropospheric warm advection and latent heat release over the mountains to the west of the UIB, due to the higher moisture content, can enhance the wind circulation at $700\,\mathrm{hPa}$ so that $W700$ remains higher than in the general case ($4.4\,\mathrm{m\,s^{-1}}$ instead
of 4.1) despite the weaker geopotential anomaly at $300\,\mathrm{hPa}$ (Figure 12-B and D).



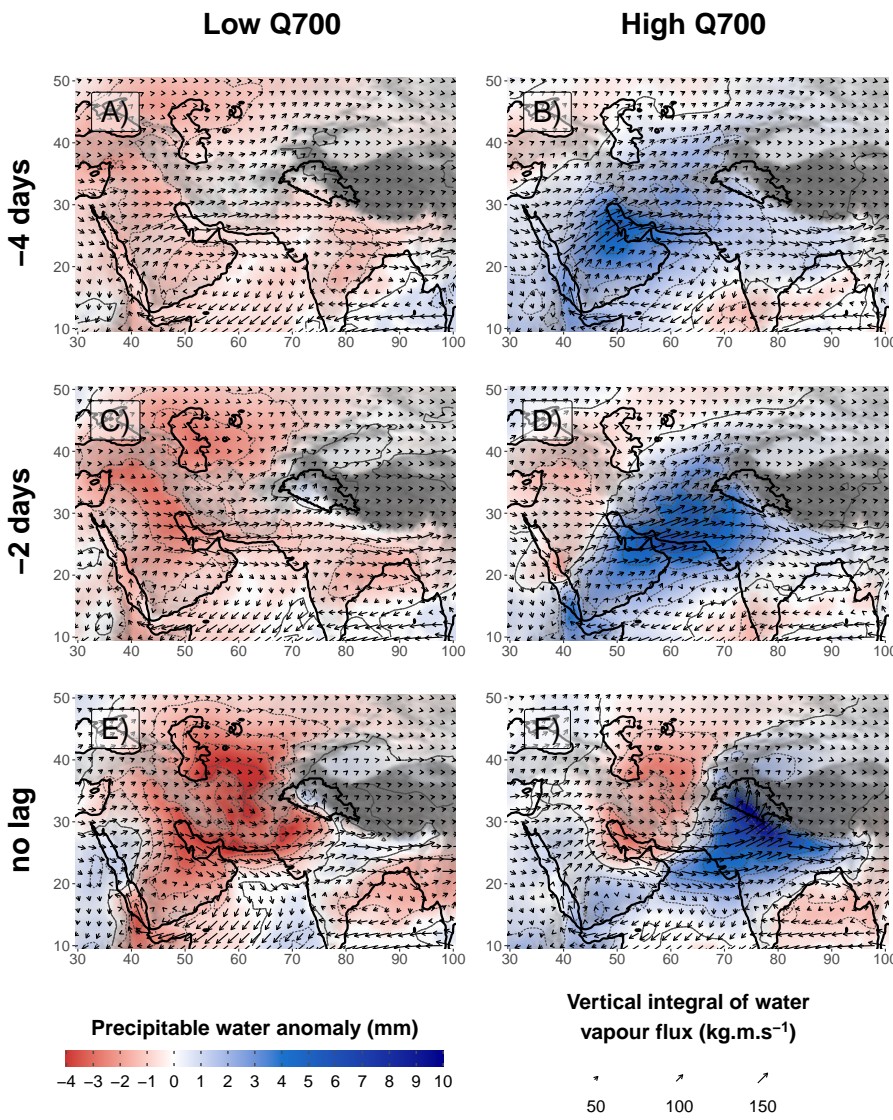

**Figure 11.** Same as Figure 9 (B, C and E), but the composites are based on the lowest (highest) values of $Q700$ in panels A, C, and E (B, D and F). See the selection definition in Section 3.3.

By contrast, in case of low $Q700$, the mean WD originates from the north-west, towards the Caspian Sea, which means it interacts less with the moisture pathway. The WD is also characterised by a deep cold core, which, despite a stronger geopotential anomaly, results in a lower $W700$ than in the general case ($3.8\,\mathrm{m\,s^{-1}}$). The WDs are also much slower than in the general case ($3.3\,\mathrm{m\,s^{-1}}$), and, as for the case of lower $W700$ discussed in Section 4.3.2, we suggest that this is due to the equatorward motion of cold air at the rear of the WD, which is reinforced here by the originally more northern position of the WD. A different way to explain the growing of the WD is through its position relative to the SWJ. The SWJ produces a jet

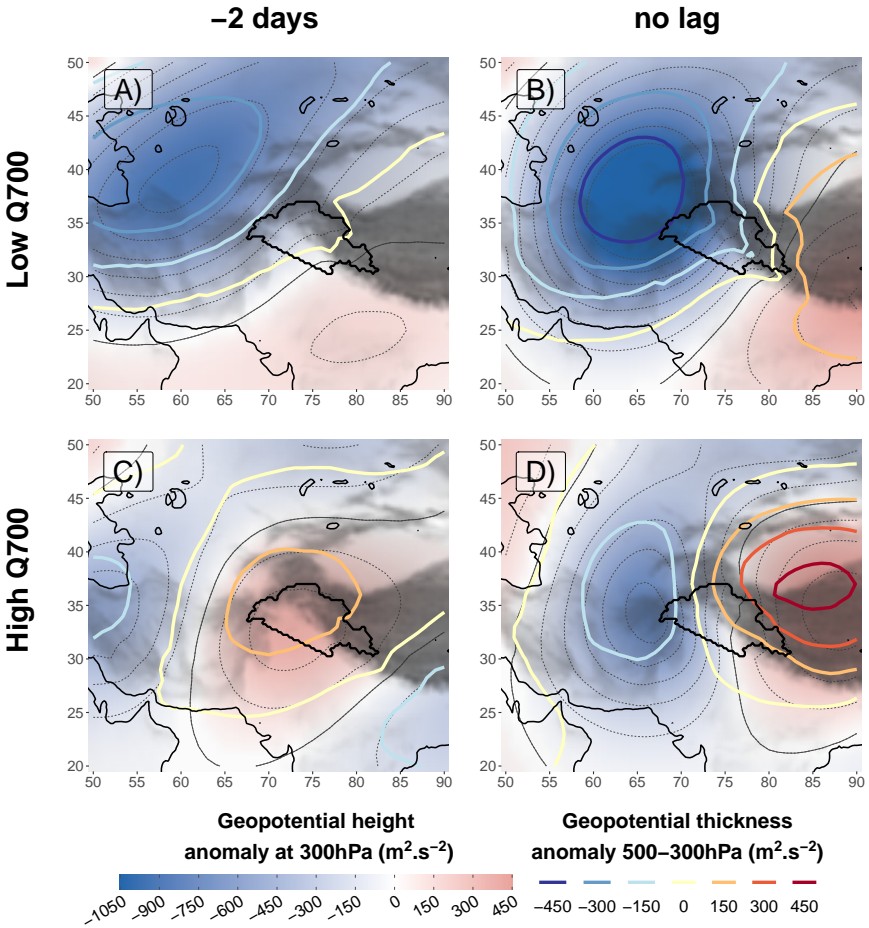

**Figure 12.** Same fields as Figure 2, but for composites based on the same selections as in Figure 11-C to F.

streak to the south-east of the WD. Consequently, such a WD is located on the left entrance of the jet, where convergence and subsidence are enhanced. Therefore, the WD can grow by stretching its vorticity. In the meantime, the subsidence leads to a further drying of the atmosphere.

Finally, moisture transport from the trade winds in the Arabian Sea also vary between the two selections: it is stronger (weaker) in case of lower (higher) $Q700$, regardless of the position of the WD (arrows, Figure 11). We discussed with Figure 9 how trade winds are impacted by the passing of a WD, but trade winds can also impact the moisture field outside of the WD influence. In particular, stronger trade winds transport the moisture evaporating from the Arabian Sea to the Southern Hemisphere, while weaker trade winds allow for a build-up of moisture. Hence, moisture supply to the IUB is also coupled to
the tropical and sub-tropical variability.

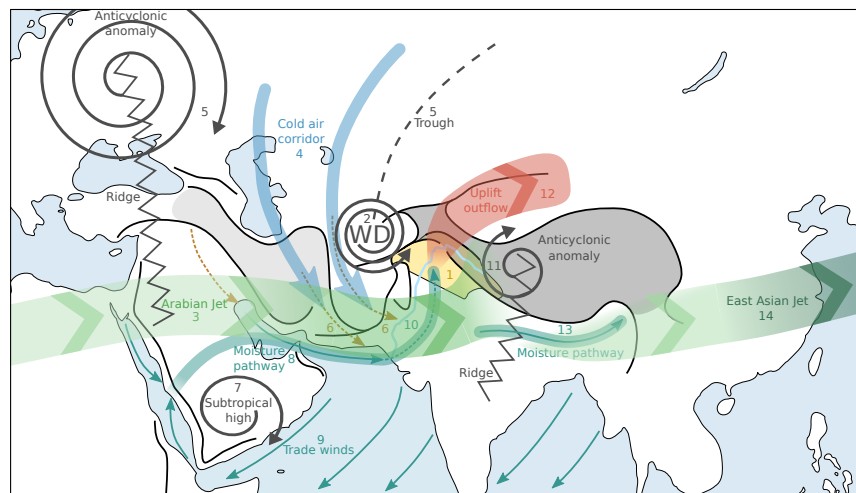

**Figure 13.** Sketch summarising the atmospheric circulation when a WD interacts with the relief to produce precipitation in the UIB. The UIB is in yellow, and the Indus River in pale blue. The black lines indicate major mountain ranges, and the grey-shaded areas high plateaus (The Tibetan Plateau and the Iranian Plateau). The spirals represent tropospheric geopotential anomalies or centres of actions. The thin sea-green arrows are mean lower troposphere moisture transport. A thicker accent indicates higher altitude transport, and a dotted line indicates transient transport due to the WD. The brown dotted lines indicate advection of dry air by the WD near the surface. The blue arrows are troposphere wide cold air advections. The thick green arrows represent the upper troposphere jets, with darker colour indicating stronger winds. The red arrow relates to the warm upper troposphere outflow. Finally, the numbers are in-text references.

## 6  Summary

This study has developed an in-depth analysis of the synoptic variability of precipitation in the Upper Indus Basin (UIB) during winter using PC regressions, quantile regressions and composites, furthering methods developed in Baudouin et al. (2020a). Several processes explaining WD growth, decay, interaction with the relief, and relationship with precipitation are suggested.

In particular, the atmospheric circulation related to precipitation has been discussed and is summarised in Figure 13.

Precipitation in the UIB (in yellow, 1) is related to the arrival of an upper troposphere disturbance from the west (Western Disturbances, 2). This disturbance is embedded in the Subtropical Westerly Jet, which forms to the west a local maxima, the Arabian Jet (3). The upper disturbance grows in part from the equatorward advection of cold air through baroclinic processes (4). This cold air is funnelled between an anticyclonic anomaly and a trough (5) and is responsible for the strengthening the

Arabian Jet (3). The cold air advection has an ambivalent role as it is accompanied by dry air close to the surface that can suppress precipitation (6).

Winter moisture transport is driven by the Arabian subtropical high (7). North of it, a westerly moisture pathway (8) connects the Red Sea with the Persian Gulf and the north of the Arabian Sea, while trade winds (9) blow over the rest of the Arabian Sea. The passing of a WD enhances the westerly moisture pathway (8), while it weakens trade winds (9). As the WD approaches





the UIB, it steers the moisture across the Indus Plain into the notch formed by the relief (10). The presence of an anticyclonic anomaly east of the WD further enhances the meridional transport (11). This advection of warm and moist air sustains itself, in particular its component close to the surface, through baroclinic and latent heat feedbacks at the synoptic scale.

Eventually, the moisture converges at low-level (around $700\,\text{hPa}$) in the UIB, over the foothills, where the uplift triggers condensation and precipitation (1). It also allows for a mixing and a warming of the air column in the UIB, which counter-
balances the cold air advection associated with the WD (4). In other words, the uplift advects low potential vorticity values to high altitude ($250\,\text{hPa}$) as indicated by the swift anticyclonic turn of its outflow (12), which then increases the zonal gradient of potential vorticity. Consequently, the upper troposphere cyclonic circulation of the WD further propagates downward, while the baroclinic uplift and the cross-barrier moisture transport increases (10). The increased vertical velocities and high altitude convergence lead to a break in the Arabian jet (3).

This positive baroclinic interaction, however, is short-lived. Once the WD reaches the UIB, its high PV anomaly is consumed by the continued advection of low PV, which quickly leads to the weakening of the cyclonic circulation. Furthermore, in addition to this negative baroclinic feedback, dry continental air intrudes into the Indus Plain (6) and cuts off the moisture supply to the UIB, which effectively further suppresses precipitation. The remaining moisture surplus is then pushed towards northeast India (13), while trade winds in the Arabian Sea re-intensify (9). Finally, as the result of the WD interaction with the
relief, the Subtropical Westerly Jet weakens to the east of the UIB and this negative anomaly propagates along the East Asian Jet in the following days (14).

Winter precipitation and cross-barrier wind in the UIB is unequivocally related to the passing of Western Disturbances close to the tropopause. However, the idealised scenario presented in Figure 13 hides large variability in the different features discussed. On the one hand, a PC regression demonstrated that cross-barrier wind is not only affected by how deep the upper
geopotential anomaly of the WD is but also by the geopotential gradient east of the WD and by the position and intensity of the cold core of the WD. On the other hand, a quantile regression showed that the build up of moisture in the UIB is dependent on the history of the WD's development and track. The importance of these effects is reinforced by the approximately quadratic relationship that exists between precipitation and cross-barrier moisture transport at $700\,\text{hPa}$ and between cross-barrier wind and WD characteristics. The latter may also be related to non quasi-geostrophic and meso-scale effects (e.g. convection or
frontal activity), which haven't been explored here but could constitute a topic for future investigations. Other topics whose analyses can build upon the methods and results presented here include the origin of Western Disturbances and their relation to anticyclonic anomalies over Eastern Europe, the seasonal cycle of precipitation, and its intra-seasonal and inter-annual variability, all of which have been only partially addressed in (e.g. Baudouin, 2020).

*Code availability.*  The code in R used to produce the figures is available at https://doi.org/10.5281/zenodo.5115561

*Author contributions.*  Original idea, analysis and text by JPB, guidance and review by MH and CP



*Competing interests.* The authors declare that they have no conflict of interest

*Disclaimer.* This paper consists in large part of the Chapter 4 of the thesis of Jean-Philippe Baudouin, submitted for the degree of Doctor of Philosophy at the University of Cambridge (https://doi.org/10.17863/CAM.68377).

*Acknowledgements.* We first acknowledge the ECMWF teams in charge of producing the ERA5 reanalysis and sustaining the online avail-
ability of the datasets. We also acknowledge our editor and our reviewers (to be precised).

JPB thanks the two examiners of his PhD dissertation, Andrew G. Turner and Andrew N. Ross, for their constructive discussions that helped shaping this paper. JPB also personally thanks Kira Rehfeld, the STACY group, and the *PalMod-Phase 2* project funded by the German Federal Ministry of Education and Research (BMBF), for their support towards the completion of his PhD.

This research was carried out as part of the *TwoRains* project, which is supported by funding from the European Research Council (ERC)
under the European Union's Horizon 2020 research and innovation programme.

For the financial support section: ERC grant agreement no. 648609, and BMBF grant no. 01LP1926C





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
