# Peer review of "Synoptic processes of winter precipitation in the Upper Indus Basin"

_Weather and Climate Dynamics, 2021_

## Author Response (AR1)

**Answers to 1ˢᵗ reviewer**

Comment: "I strongly think 'cross-barrier' term to be replaced with orographic interaction in abstract and also later if there in manuscript too where ever it is meant for that purpose."

Answer: We agree that precipitation are triggered by "orographic interaction", however, we found this expression too generic as it includes all type of interaction between a mountain and the atmospheric flow. Here, we are specifically talking about the forced uplift of a moisture flow directed towards a mountain range, hence "cross-barrier moisture flow". In the text "cross-barrier" is always an adjective associated with "wind", "flow" or "moisture transport", and we don't see how " orographic interaction" would have the same meaning. In fact, "cross-barrier flow" is a relatively common expression in the literature on mountain meteorology (e.g. Mountain Meteorology: Fundamentals and Applications, Whiteman 2000, Mountain Weather Research and Forecasting, Bradley et al. 2021)

Changes:

l1. "is triggered by cross-barrier moisture transport" → is triggered by orographic interaction and the forced uplift of a cross-barrier moisture flow.

l. 17 "precipitation in the UIB is triggered by the forced up-lift of a moisture transport perpendicular to the mountain ranges" → most of the precipitation in the UIB is triggered by orographic interaction, and more specifically, by the forced up-lift of a moisture transport perpendicular to the mountain ranges.

Comment: "Now the big question: all winter, if DJF, precipitation is not always by WDs. There is precipitation during to non-WDs days as well. In addition, there are WDs but at times they all don't precipitated. Authors are advised to distinctly make this clear."

Answer: You are right, we should have tackled this question in the conclusion. we also made minor changes to the introduction. However, "WD-days" is very dependent on the definition of a WD: e.g. location of maximum of vorticity, and associated threshold. Without these strict definitions, we showed here is that 1) the gradient over the IUB is more important than the location of the WD center, 2) the thermal structure/baroclinicity greatly modulates the strength of a WD. The two regressions (Fig1b and Fig5a) show an impressive good match that can only indicate that all the main processes leading to precipitation in winter have been considered. Finally, this work could be used to design better WD indices or WD definitions.

Changes:

l. 26 "Winter precipitation events in the UIB are related to the passing of extra-tropical, synoptic-scale, disturbances" → Winter precipitation events in the UIB are in general related to the passing of extra-tropical, synoptic-scale, disturbances

l. 569-575: On the one hand, a PC regression demonstrated that cross-barrier wind is not only affected by how deep the upper geopotential anomaly of the WD is but also by the geopotential gradient east of the WD and by the position and intensity of the cold core of the WD. These

characteristics could be used to explain why some WDs trigger very little precipitation. On the other hand, a quantile regression showed that the build up of moisture in the UIB is dependent on the history of the WD's development and track. The importance of these effects is reinforced by the approximately quadratic relationship that exists between precipitation and cross-barrier moisture transport at \SI{700}hPa and between cross-barrier wind and WD characteristics. The latter may also be related to non quasi-geostrophic and meso-scale effects, potentially unrelated to WDs (e.g. convection or frontal activity), which haven't been explored here but could constitute a topic for future investigations.

Comment: "And I advise authors for a future study to take pressure; temp-moisture feedback mechanisms; vorticity together to determine that. As authors themselves are as well mentioning 'baroclinicity'."

Answer: we don't understand the comment and what "that" is referring to. And we do consider geopotential (better than sea level pressure in this context of heterogeneous terrain), temperature-moisture feedback and vorticity in this study.

Comment: "And, to include impacts of gravity/Kelvin/Rossby waves too."

Answer: we do not see how gravity and Kelvin waves are relevant for precipitation in the UIB, but that may be due to a lack of knowledge on my part. As for Rossby waves, they are of utmost importance for the propagation of WDs, and particularly when investigating teleconnections.

Comment: "Don't agree with statement 32-24: 'Despite the abundant interest, a precise and non case-specific understanding of the relationship between WD characteristics and precipitation variability is still lacking'. I would appreciate if authors reframe this sentence with mention of 'physical processes are still less understood'."

Changes:

l. 31-32: "Despite the abundant interest, a precise and non case-specific understanding of the relationship between WD characteristics and precipitation variability is still lacking." → Despite the abundant interest, the general physical processes explaining the relationship between WD characteristics and precipitation variability are not well enough understood.

Comment: "In Fig. 1 I will suggest to include map of Indian subcontinent first as (a) so that most of the readers can geographically know the study region."

Changes: A new figure has been added, representing the relief of South Asia and the area of interest (Upper Indus Basin)

Comment: "Sections 2 and 3 are perfectly fine in detailing the study region, data, methods etc. I still have reservation in using term 'cross-barrier', In fact this orographic interaction is the process which modulates the precipitation forming mechanisms."

Answer: cf. above

Comment: "What is the rationale of mentioning it as cold core? Either provide or rename something else. Figure 3 (no lag) suggest it is warm core."

Answer: That's a key characteristic of the WDs, by opposition to the warm-core in tropical cyclone for example. The section itself is the rationale. Figure 3 clearly shows a cold core in the troposphere below the minimum of geopotential anomaly. There is indeed a warm anomaly in the lower stratosphere by adjustment, but that's not what we are interested in.

Changes:

l203: "The cold core" → "The tropospheric cold core"

Comment: "There is need of showing orgaphic interaction: as upslope flow along the windaord side and then sinking of the flow, ifonce cross the barrier."

Answer: To me this is just paraphrasing the sentence in the second paragraph of the introduction (l.23-24): "precipitation in the UIB is triggered by the forced up-lift of a moisture transport perpendicular to the mountain ranges". Sinking is not particularly important for the precipitation, although we do discuss in the result section. We think the reviewer might have understood cross-barrier wind as through the range, without changing altitude, which isn't physically possible.

Comment: "Pls see work: western Disturbances: A review."

Answer: This paper is cited 13 times.

**Answers to 2[nd] reviewer**

L37. "Convergence is one of several necessary ingredients for heavy rainfall (see, e.g., Doswell et al 1996, doi:10.1175/1520-0434(1996)011<0560:FFFAIB>2.0.CO;2). Convergence alone is insufficient to trigger precipitation."

Answer: We agree with the reviewer, but the sentence doesn't suggest that convergence is only the ingredient for precipitation, it merely discusses the different of altitude between maximum convergence and maximum cyclonic circulation. Overall, the study focuses indeed on moisture convergence, at least implicitly, through cross-barrier moisture transport. The result are pretty convincing (cf. Fig1b) and yet the limitations discussed (in particular with respect to he quadratic fit)

L79. "Is this minimum geopotential over the whole dataset length?"

Change: "grid point where their all-time minimum geopotential is above the model surface were deselected"

L95. "46 PCs seems like an arbitrary truncation, so presumably there must've been some reasoning behind it?"

Change: A new footnote has been added:

"This number insures the "extent of pattern" as defined in Baudouin et al. (2020a) is the same, see also Baudouin (2020)"

L. 105 "I wonder whether something like "UV700" would be better here. A reader scanning through quickly may confuse "W700" for vertical wind speed."

We agree, W700 has been replaced by UV700

L109 "Is "Q700" weighted specific humidity (i.e. in accordance with the units of Fig 1B)? You don't seem to state it in the text."

Change: "For the moisture contribution (labelled hereafter $Q700$), the time series of specific humidity at each location are weighted with the euclidean norm of the coefficients of both meridional and zonal moisture transport" (l. 107-108)

Sec 3.3 "I had some difficulty following the methodology in Sec 3.3. I'm not entirely convinced an interested reader could reconstruct the work based on this text."

Change: The text in this section has (hopefully) been clarified:

"There is variability in the dynamic structure of WDs. To investigate this variance, a more complex composite analysis is used, based on two quantile regressions. To maintain comparability with the other composite analysis (cf. Section 3.2), the regressions are performed using the timesteps whose value of UV700 is above the 90$^{th}$ percentile . The regressions predict respectively the first and third quartile of Hat_UV700. Hat_UV700 (see Equation 1) is preferred to UV700 itself, so that the differences between the two subsets are not related to the variability missed by the PC regression of UV700 with geopotential heights and thicknesses. We regress the values of Hat_UV700 at the selected timesteps on several predictors: Z300, geopotential anomalyat300 hPa at the centre of the mean WD (66∘E - 36∘N), and months. Two subsets are eventually created that include all the timesteps whose value of Hat_UV700 is conditionally below the first and above the third quartile respectively. These two subsets are used as composite and hereafter referred to as "Lower UV700" and "Higher UV700". The properties of the quantile regression ensure that the two composites have the same average  characteristics  regarding the predictors (or conditions), but a mean value off Hat_UV700 as different as possible. Hence, the inclusion of Z300 and geopotential anomaly as predictors guarantee that the composite WD in each subset is located at a similar place and has a similar intensity. Months are also included to avoid seasonal biases regarding WD characteristics: for each month, the same number of timesteps is present in each subset. Figures 7 and 8 represent the same variables as in Figures 2 and 4, respectively but are based on these new composites. Finally, a second sampling is performed to investigate the specific humidity variability in WDs. Two new subsets (hereafter"Low Q700" and "High Q700") are computed based on two quantile regressions respectively predicting the first and third quartile of Q700. We regress the values of UV700 at the same timesteps as above. The predictors are the months, so as to remove the impact of seasonality, and the difference of the geopotential height anomaly between the grid points 60∘E-36∘N and 70∘E-36∘N. This latter predictor fixes the longitudinal gradient of geopotential across the UIB, and in this way, the position of the composite WD. However, the intensity of the WDs is not fixed as in the previous case. Various composites are derived from these selections: Figures 11, and 12, respectively comparable to Figures 9 and 2.

L230-235 "I completely agree; I've long thought the same (e.g. Fig 4ai of doi: 10.1002/qj.3200) as it makes more sense when one considers the dynamics at play."

Thank you for your support!

L295 "Not quite sure about the wording here, a gradient in geopotential doesn't trigger advection since they are linked through a diagnostic equation with no time dependence. Consider revising."

Change: "This gradient is indicative of a southerly advection"

L303 "PC regression indicates that the presence of an east-west geopotential gradient across the UIB is more important than the proximity of a WD" – This is perhaps phrased in a slightly misleading way since gradient in geopotential is caused by a WD being slightly upstream, so the location of the WD is still important. Also slightly vague as what it is important for is not stated."

Answer: We didn't say the "location" but the "proximity" of the WD. What we mean is that it doesn't matter much in terms of cross-barrier wind whether the WD is 100km or 3000km away from UIB, as long as it drives the proper geopotential gradient.

Change: "is more important ==for the cross-barrier wind== than the proximity of a WD==, as long as it remains upstream of the UIB=="

L305-315 "Do the authors think that this strong anticyclonic anomaly is in turn associated with a jet streak? The dynamics associated with a jet streak entrance may explain the larger values of W700 and precipitation in this composite. I see the authors touch on this idea in L380."

Answer: This anticyclonic anomaly is indeed both important to increase the geostrophic cross-barrier wind at altitude, and the baroclinic uplift.

L324 "The direction of causality in this statement needs additional evidence if the authors intended it to read that way. Otherwise, consider rewording."

Change: "The speed of the WD also varies depending on the peak intensity of W700" → "The speed of the WD is also different whether higher or lower W700 is considered"

L 329-334 "I'm not sure I follow the reasoning here. If the WD is not moving faster due to advection, as the authors state, then there must be some source of additional vorticity generation downstream. Are the authors suggesting that this is generated baroclinically through WAA?"

Answer: Yes, but it's rather cold air advection ("the equatorward motion of cold air at the rear of the WD"). Because we are looking at the top of the troposphere, it's the reverse than what is happening at the surface, with respect to tropospheric temperature anomalies.

L351 "How does the zonal temperature gradient at 300 hPa affect the geopotential gradient at 700 hPa? The authors have already demonstrated substantial variability in thickness between these levels."

Answer: We actually meant the tropospheric zonal temperature gradient, not specifically at 300hPa (Changed in text). In fact, we are arguing that these temperature effects directly impact the thickness.

L368 "I would argue that these views could be seen as complementary rather than contrasting, but then again, I am the lead author of the cited study!"

Change: "We ==further== suggest"

Fig9. "the vector field has a very high resolution, which makes the pattern difficult to see unless zoomed in quite far. Consider coarsening."

Answer: We hope that the figure takes a full page on the printed version! We found the details in the UIB quite interesting, to clearly see the rotation and acceleration of the flow towards the UIB.

L435 "This a sensible conclusion, and increasingly supported by isotopic studies of precipitation over the UIB (e.g. Jeelani et al, 2017, doi:10.1007/s12040-017-0894-z; Dar et al, 2021, doi:10.1029/2020JD032853)."

Answer: The two papers have been cited

L528: "Are the authors sure that vortex stretching likely to be occurring here? On the left side of a jet streak entrance region, there is vorticity creation aloft, but it is balanced by destruction below."

Answer: It is important to distinguish what is happening at the surface, and at the tropopause level, where the WD is located. So, yes, "there is vorticity creation aloft", which helps the WD to grow at altitude, but that doesn't translate into an increased vorticity close the surface, quite the contrary. And that explains why the WD appears to be more intense at altitude, but doesn't increase the low level convergence.

"Therefore, the WD can grow by stretching its vorticity, while the surface cyclonic circulation and the convergence at \SI{700}hPa is reduced."

Other Comments:

L89 "By "wind" do you mean "wind speed"?"

Change: "defined as the product of wind (both components) and specific humidity"

L127: These are units of geopotential, not geopotential height.

Answer: We agree, we have been mixing the two terms in the entire paper. It has been corrected to "geopotential"